# Advancing Text-to-3D Generation with Linearized Lookahead Variational Score Distillation

## Abstract

Text-to-3D generation based on score distillation of pre-trained 2D diffusion models has gained increasing interest, with variational score distillation (VSD) as a remarkable example. VSD proves that vanilla score distillation can be improved by introducing an extra score-based model, which characterizes the distribution of images rendered from 3D models, to correct the distillation gradient. Despite the theoretical foundations, VSD, in practice, is likely to suffer from slow and sometimes ill-posed convergence. In this paper, we perform an in-depth investigation of the interplay between the introduced score model and the 3D model, and find that there exists a mismatching problem between LoRA and 3D distributions in practical implementation. We can simply adjust their optimization order to improve the generation quality. By doing so, the score model looks ahead to the current 3D state and hence yields more reasonable corrections. Nevertheless, naive lookahead VSD may suffer from unstable training in practice due to the potential over-fitting. To address this, we propose to use a linearized variant of the model for score distillation, giving rise to the Linearized Lookahead Variational Score Distillation ($L^2$-VSD). $L^2$-VSD can be realized efficiently with forward-mode autodiff functionalities of existing deep learning libraries. Extensive experiments validate the efficacy of $L^2$-VSD, revealing its clear superiority over prior score distillation-based methods. We also show that our method can be seamlessly incorporated into any other VSD-based text-to-3D framework.

## 1 Introduction

3D content creation is important for a variety of applications, such as interactive gaming (Bruce et al., 2024; Xia et al., 2024), cinematic arts (Conlen et al., 2023), AR/VR (Creed et al., 2023; Li et al., 2024), and building simulated environments for training agents in robotics (Team et al., 2024). However, it is still challenging and expensive to create a high-quality 3D asset as it requires a high level of expertise. Therefore, automating this process with generative models has become an important problem (Jiang, 2024), while remaining non-trivial due to the scarcity of training data and the complexity of 3D representations.

Score distillation has emerged as an attractive way for 3D generation given textual condition (Poole et al., 2022; Lin et al., 2023; Chen et al., 2023; Wang et al., 2023; 2024a). It leverages pretrained 2D diffusion models (Ho et al., 2020; Rombach et al., 2022) to define priors to guide the evolvement of 3D content without reliance on annotations. Score Distillation Sampling (SDS) (Poole et al., 2022) is a seminal work in this line, but it is widely criticized that its generations suffer from the over-smoothing issue. Variational Score Distillation (VSD) (Wang et al., 2023) remediates this by introducing an extra model that captures the score of the images rendered from the 3D model to correct the distillation gradient. However, VSD often requires a lengthy optimization through 3 stages: NeRF generation, geometry refinement, and texture refinement. The outcomes obtained in the initial stage are often blurry, prone to collapsing, and not directly applicable (Wei et al., 2023). Though existing works begin to understand and improve SDS (Wang et al., 2024a; Yu et al., 2023; Katzir et al., 2023), there are great but less efforts dedicated to improving the more promising VSD (Ma et al., 2025; Wei et al., 2024).

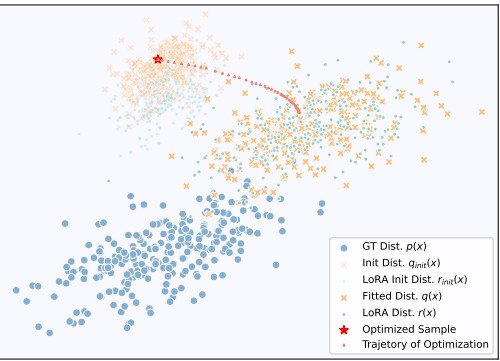 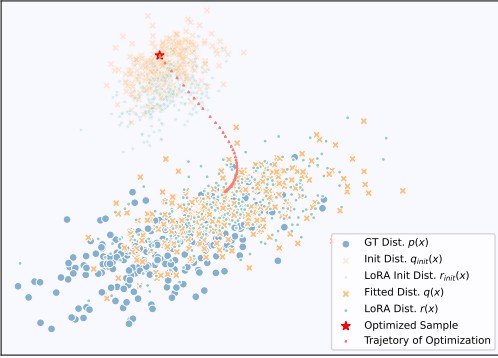

Figure 1: **Comparison of convergence between VSD and L-VSD with an illustrative 2D Gaussian example.** In this toy example we consider $x \in \mathbb{R}^2$ from a single Gaussian distribution. We optimize $q(x)$ towards ground truth distribution $p(x)$ and use $r(x)$ to function as LoRA which is used to estimate $q(x)$. This example validates the existence of mismatching issue in VSD and we leave the details in Sec. 3.2.

To identify the root of VSD's drawback, we conduct a comprehensive analysis of the interaction between the introduced score model and 3D model revealing that adjusting their optimization order can sometimes lead to a considerable enhancement in generation quality. This adjustment allows the score model to look ahead to the current 3D state, resulting in more accurate and sensible corrections for the distillation gradient. Yet, naive lookahead VSD can encounter unstable training due to the risk of the score model overfitting the single 3D particle and the sampled camera view.

To address this issue, we formally compare the correction gradients before and after looking ahead and identify two major differences—a linear first-order term and a high-order one. Upon closer examination, we observe that the former accommodates subtle semantic information, whereas the latter contains non-trivial high-frequency noises. Given these findings, we propose to use only the first-order term for correction, yielding *Linearized Lookahead Variational Score Distillation* ($L^2$-VSD), to reliably and consistently boost the generation quality of VSD. $L^2$-VSD is both easy to implement and computationally efficient—the added linear term can be computed by only one additional forward process of the score model under the scope of forward-mode automatic differentiation, which is supported in many deep learning libraries (Paszke et al., 2017; Ketkar et al., 2021).

Through extensive experiments, we demonstrate the significant superiority of the proposed $L^2$-VSD in improving 3D generation quality compared to competing baseline methods, as shown in Fig. 7. $L^2$-VSD can even produce realistic generation results with low resolution directly in the first stage. Moreover, we empirically show that $L^2$-VSD can be seamlessly integrated into other VSD-based 3D generation pipelines and combined with other parallel techniques for VSD, e.g., Entropy Score Distillation(ESD) (Wang et al., 2024a) which mitigates the Janus problem, for further improvement.

We summarize our technical contributions as follows:

- For VSD, a fundamental method in text-to-3D generation, we carefully identify the gaps between its theory and implementation and analyze the potential impact the gaps may bring us, providing a direction for possible improvements.
- We propose $L^2$-VSD, an easy to implement and computationally efficient variant of VSD, which mitigates the mismatching problem to some extent. We demonstrate its significant improvement over baselines and comparable to other state-of-the-arts.
- We demonstrate that our method can be seamlessly combined with other VSD-based improving techniques without much effort.

## 2 BACKGROUND

### 2.1 DIFFUSION MODELS

Diffusion models (DMs) (Sohl-Dickstein et al., 2015; Ho et al., 2020; Song et al., 2020) are defined with a forward diffusion process on data $x \in \mathbb{R}^d$ with a Gaussian transition kernel. The conditional distribution at some timestep $t \in [0, T]$ usually satisfies

$$q(x_t|x_0) = \mathcal{N}(x_t; \alpha_t x_0, \sigma_t^2 \mathbf{I}) \tag{1}$$

where $x_0 := x$, and $\alpha_t$ and $\sigma_t$ are pre-defined noise schedules. DMs learn a reverse diffusion process, specified by a parameterized distribution $p(x_{t-1}|x_t) := \mathcal{N}(x_{t-1}; \mu_\psi(x_t, t), \sigma_t^2 \mathbf{I})$ with $\mu_\psi$ as a neural network (NN), to enable the sampling of generations from Gaussian noise. The training objective of $\mu_\psi$ is the variational lower bound of the log data likelihood. In practice, $\mu_\psi$ is re-parameterized as a denoising network $\epsilon_\psi$ and the training loss can be further simplified as a Mean Squared Error (MSE) form (Ho et al., 2020; Kingma et al., 2023):

$$\mathcal{L}_{Diff} := \mathbb{E}_{x,t,\epsilon}[\omega(t)||\epsilon_\psi(\alpha_t x + \sigma_t \epsilon, t) - \epsilon||_2^2], \qquad (2)$$

where $x$ follows the data distribution, $t$ is uniformly drawn from $[0, T]$, $\epsilon$ is a standard Gaussian noise, and $\omega(t)$ is a time-dependent coefficient. $\epsilon_\psi$ also inherently connects to score matching (Vincent, 2011; Song et al., 2020).

**Classifier-free guidance (CFG)** (Ho & Salimans, 2022). We can augment the model $\epsilon_\psi$ with an extra input, the condition $y$, to characterize the corresponding conditional distribution, leaving $\epsilon_\psi(x_t, t, \emptyset)$ account for the original unconditional one. Then, we can resort to CFG to further boost the quality of conditional generation. Typically, CFG leverages the following term in the sampling process:

$$\hat{\epsilon}_\psi(x_t, t, y) := (1 + s)\epsilon_\psi(x_t, t, y) - s\epsilon_\psi(x_t, t, \emptyset) \qquad (3)$$

where $s > 0$ refers to a guidance scale.

## 2.2 TEXT-TO-3D GENERATION WITH SCORE DISTILLATION

Text-to-3D generation aims to identify the parameters $\theta \in \mathbb{R}^N$ of a 3D model given a text condition $y$. Neural radiance field (NeRF) (Mildenhall et al., 2020) is a typical 3D representation based on neural networks. In particular, NeRF renders a new view of the scene with the input of a sequence of images as known views. Additionally, textured mesh can be applied to represent the geometry of a 3D object with triangle meshes and textures with color on the mesh surface.

Denote $g(\theta, c)$ as the differential rendering function projecting the 3D scene to a 2D image given a camera angle $c$. Score distillation approaches for text-to-3D generations demand the image sample $g(\theta, c)$ to respect the prior specified by a text-to-2D diffusion model $\epsilon_{pretrain}(\cdot, \cdot, y)$ pretrained on vast real text-image pairs, based on which the optimization goal is constructed (Poole et al., 2022; Wang et al., 2023; 2024a; Yu et al., 2023; Tang et al., 2024; Yang et al., 2023; Katzir et al., 2023).

**Score Distillation Sampling (SDS)** (Poole et al., 2022) updates the 3D model using view-dependent prompt $y^c$:

$$\nabla_\theta \mathcal{L}_{SDS}(\theta) := \mathbb{E}_{t,\epsilon,c}[\omega(t)(\epsilon_{pretrain}(x_t, t, y^c) - \epsilon)\frac{\partial g(\theta, c)}{\partial \theta}], \qquad (4)$$

where $c$ is a randomly sampled camera angle and $x_t := \alpha_t g(\theta, c) + \sigma_t \epsilon$. The gradient is a simplification of that of the denoising objective w.r.t. $\theta$ (Poole et al., 2022). Intuitively, it encourages the rendered images to move toward the high-probability regions of the pretrained model, thus a good 3D model emerges. SDS is a seminal work in the line of text-to-3D generation, but it is sensitive to the CFG scale $s$ (Ho & Salimans, 2022). A small $s$ often results in over-smooth outcomes, whereas a large $s$ leads to over-saturation.

**Variational Score Distillation (VSD)** (Wang et al., 2023) addresses the issue of SDS with a thorough theoretical analysis and proposes a new algorithm. In particular, apart from the pretrained model $\epsilon_{pretrain}(x_t, t, y)$ for capturing the data distribution, VSD introduces a tunable model $\epsilon_\phi(x_t, t, c, y)$, often instantiated as a LoRA (Hu et al., 2021) adaptation of $\epsilon_{pretrain}$, to account for the distribution of images rendered from all possible 3D models given condition $y$ and camera angle $c$. We refer the introduced model as the LoRA model hereafter. VSD proves the LoRA model can reliably correct the original distillation gradient.

VSD, in practice, usually considers only a single 3D particle $\theta$ and performs an iterative optimization of $\theta$ and $\phi$ until convergence. More specifically, let $\theta_i$ and $\phi_i$ denote the parameters at $i$-th training iteration. VSD updates $\theta_i$ with the following gradient:

$$\nabla_{\theta_i} \mathcal{L}_{VSD}(\theta_i) := \mathbb{E}_{t,\epsilon,c}\left[w(t)(\epsilon_{pretrain}(x_t, t, y) - \epsilon_{\phi_i}(x_t, t, c, y))\frac{\partial g(\theta, c)}{\partial \theta}\Big|_{\theta=\theta_i}\right], \qquad (5)$$

where $x_t = \alpha_t g(\theta_i, c) + \sigma_t \epsilon$. To avoid more than once differentiable rendering of the 3D model for efficiency, VSD updates $\phi_i$ still with $g(\theta_i, c)$ while using a different noisy state $x_{t'} = \alpha_{t'} g(\theta_i, c) +$

(a) **VSD:** From left to right, the results correspond to setting $\gamma$ to 1, 2, 5, and 10. We can observe that the quality of the model is not fundamentally improved.

(b) **L-VSD:** From left to right, the first three results correspond to setting $\gamma$ to 1, 2, and 5; the last corresponds to scaling the learning rate by 0.1 with $\gamma = 1$.

Figure 2: **Qualitative Examples of training LoRA model for multiple steps per optimization iteration for VSD and L-VSD.**

$\sigma_{t'}\epsilon'$. The learning objective is the denoising loss defined in Equation (2), whose gradient is:

$$\nabla_{\phi_i}\mathcal{L}_{VSD}(\phi_i) := \mathbb{E}_{t',\epsilon',c}\left[(\epsilon_{\phi_i}(x_{t'},t',c,y) - \epsilon')J_{\phi_i}(x_{t'},t',c,y)\right]. \tag{6}$$

where $J_{\phi_i}(x_{t'},t',c,y) := \frac{\partial \epsilon_\phi(x_{t'},t',c,y)}{\partial \phi}|_{\phi=\phi_i}$ and we omit some time-dependent scaling factor.

Apart from the compromise to maintain one 3D particle for efficiency, the practical algorithm of VSD exhibits several significant gaps from the theory: 1) one-step update cannot guarantee $\phi$ to converge in each iteration, and 2) the updates to $\theta_i$ are computed given the LoRA model $\phi_i$, which characterizes the distribution associated with the previous 3D state $\theta_{i-1}$ instead of $\theta_i$. We hypothesize these are probably the root of the unstable performance and sometimes corrupted outcomes of VSD, and perform an in-depth investigation regarding them below.

## 3 DIAGNOSE THE ISSUES OF VSD

To assess whether the identified gaps contribute to the issues of VSD, we conduct two sets of experiments in this section. As introduced in Sec 2.2, we analyze the impact of convergence of LoRA in Sec 3.1; then, we correct the optimization order to see the consequences in Sec 3.2; lastly, we combine these two factors to check if they are interfered in Sec 3.3. We provide an illustrative 2D Gaussian example for better understanding and to evaluate the effectiveness of lookahead in Sec 3.2. We base the implementation on the open-source framework threestudio (Guo et al., 2023).

### 3.1 MAKE THE LORA MODEL BETTER CONVERGED? MAYBE NO.

**Assumption.** The one-step update in vanilla VSD cannot guarantee the LoRA model to converge well, thus possibly harming the effectiveness of score distillation. We assume updating the LoRA model for $\gamma$ ($\gamma > 1$) steps in each iteration could alleviate the pathology.

**Experiments Setting.** We evaluate the effects of various $\gamma$, including 1, 2, 5, and 10. In each step, we optimize the LoRA model with different noisy images under different views. Unless otherwise specified, we take NeRF as the default 3D representation and use a simple prompt "a delicious hamburger" in the study. We keep all other hyper-parameters the same as the vanilla VSD. Usually, the whole VSD process contains multiple stages, where in the first stage a NeRF is constructed and then the geometry and texture are refined respectively. We directly report the first-stage learning outcomes because it establishes the foundation for the following parts.

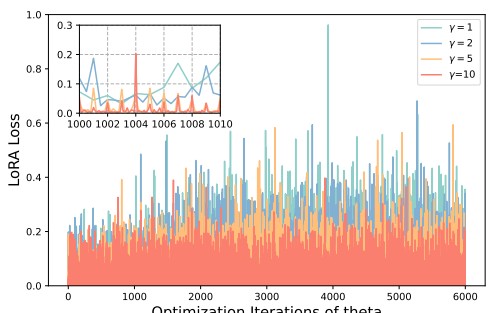

Figure 3: **Loss of the LoRA model during training given various $\gamma$.** When increasing $\gamma$, the loss is relatively lower, while also showing periodic changes.

**Results.** We present the training loss of the LoRA model in Fig. 3 to indicate the convergence and the final generations in Fig. 2a to reflect if the issues still exist. As shown, although the loss curves for various $\gamma$ share a periodic rise and fall, the loss of a larger $\gamma$ (e.g., 5 or 10) is floating in a relatively smaller range, and $\gamma = 5$ roughly makes LoRA model converge. However, as shown in Fig. 2a, it's hard to tell if the shape becomes relatively more reasonable and the overall quality of the 3D model does not witness a continual improvement as $\gamma$ rises.Thereby, we conclude that improving the convergence of the LoRA model on the original VSD is not sufficient, thus being not the key factor.

## 3.2 MAKE THE LoRA MODEL LOOKAHEAD? MAYBE YES!

**Assumption.** As shown in Equation (5), the updates to $\theta_i$ are computed given $\phi_i$, which characterizes the distribution associated with $\theta_{i-1}$ instead of $\theta_i$. This is inconsistent with Theorem 2 of (Wang et al., 2023), where the LoRA model should first adapt to the current 3D model (i.e., $\theta_i$) to serve as a reliable score estimator for the corresponding distribution. Fixing such a mismatch may address the issues of VSD.

**Experiments Setting.** We update $\phi_i$ first to obtain $\phi_{i+1}$, based on which $\theta_i$ is updated. We name such a modification **Lookahead-VSD** (**L-VSD**) because it makes the LoRA model look ahead for one step compared to the original VSD. All other parameters are kept unchanged.

**Results.** We show the result in the first column of Fig. 2b. Intuitively, the rendered image has clearer edges compared to those in Fig. 2a. Besides, inspecting the optimization process, we find L-VSD acquires the geometries and textures for the 3D model more quickly than VSD, and the loss of the 3D model in L-VSD with $\gamma = 1$ floats in a similar level to VSD with $\gamma = 10$. These results validate the necessity for the LoRA model to look ahead during the optimization of VSD. However, we also observe that the 3D model can easily suffer from being over-saturated as optimization continues, which means the 3D models can not converge with normal shapes and colors in L-VSD.

**Illustrative Example.** It seems weird and self-contradictory with only the unsatisfied results above. Here we provide an illustrative 2D Gaussian example in Fig. 1, to show the existence of mismatching problem, the effectiveness of lookahead, and point out the possible reason that harms the performance of L-VSD. In this experiment, we assume that $\theta = x \in \mathbb{R}^2$, and preset a Gaussian distribution as ground truth, which should be taken as the pretrained distribution. We randomly initialize the parameterized distribution $q(x)$, and use another Gaussian distribution $r(x)$ to approximate $q$, which can be interpreted as LoRA in VSD. In each iteration, we randomly sample $x_{sample}$ from $q(x)$, which is similar to differentiable rendering in VSD, and then use the mean and variance of Gaussian distribution to calculate the optimization direction. The learning trajectories are illustrated in Fig. 1. The results confirm the mismatching problem of VSD which hinders the distribution matching process. Correcting the optimization order can lead to better results. But, if we overfit the $r(x)$ on the samples $x_{sample}$, the results cannot converge normally towards the region of $p(x)$ when timestep is small, which usually lead to color saturation in text-to-3D generation (Huang et al., 2023; Tang et al., 2024). This evidence supports our finding with the L-VSD above. More illustrative examples are provided in Appendix. A and complete runnable code can be found in Appendix. F.

## 3.3 LOOKAHEAD × CONVERGENCE? DEFINITELY NO.

From the above studies, we learn that fitting the LoRA model to the 3D model first is essential for VSD while enhancing the convergence of the LoRA model is also beneficial to some degree. Here we conduct additional experiments to find out the consequence of combining these two factors.

Based on the setting of L-VSD in Sec. 3.2, we increase the LoRA training steps $\gamma$ to 2 and 5. We also perform a study where the learning rate of LoRA is scaled to 1/10 of the original one to indicate under-convergence. Fig. 2b exhibits the results. As shown, increasing $\gamma$ in L-VSD exacerbates the phenomenon of over-saturation and makes the geometry and texture easier to collapse, while decreasing the learning rate of the LoRA model somehow improves the quality of the generation.

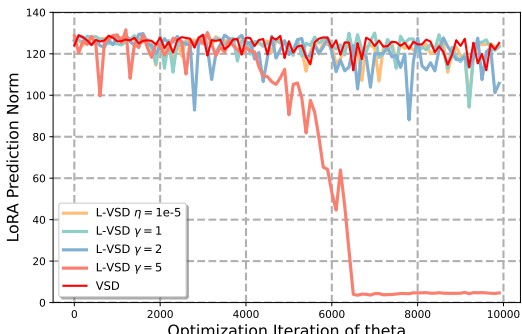

Figure 4: **The norm of the LoRA prediction $||\epsilon_\phi||$ varies w.r.t. training iteration at various $\gamma$ in L-VSD.**

These deviate from our expectations. To chase a deeper understanding, we examine the output of the LoRA by inspecting its norm $||\epsilon_\phi||$, and plot its variation during training in Fig. 4. We observe that the norm rapidly drops to zero for L-VSD with $\gamma = 5$, which implies ill-posed convergence. The underlying reason is probably that we maintain only one single 3D particle training of LoRA for efficiency, thus the distribution to fit by the LoRA model is biased. The pathology is more obvious for L-VSD when optimizing LoRA more intensively.

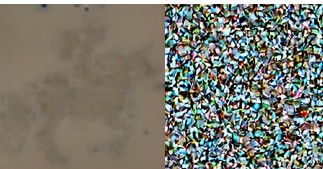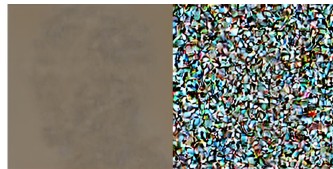

Figure 5: **Visualization of** $\Delta\epsilon_{first}$ **and** $\Delta\epsilon_{high}$**.** The left column in each group represents the decoded first-order term while the right column represents the decoded high-order term. Prompt for each group: (left) "a delicious hamburger"; (mid) "an astronaut riding a horse";(right) "an Iron man".

Besides, we note that the trick of reducing the learning rate, despite being effective in this case, is unstable when handling harder prompts. Please refer to Appendix C.1 for more failure cases.

**Conclusion** Although lookahead is important for generating 3D models with clear outlines and realistic texture, the risk is the possibility of the LoRA model over-fitting on the isolated particle, which sometimes exhibits flaws on geometry and texture after each optimization step. Given these, this work tries to figure out a way to interpolate between original VSD with relatively stable formation process and L-VSD with higher-quality outcomes.

## 4 METHODOLOGY

In Sec 4.1, we conduct a rigorous comparison between VSD and L-VSD and perform thorough studies to gain an understanding of their difference. Please also refer to Appendix A for an illustrative overview about training pipelines of VSD and L-VSD to understand their difference better. We then derive the novel Linearized Lookahead VSD ($L^2$-VSD) in Sec. 4.2.

### 4.1 COMPARE VSD WITH L-VSD

We first lay out the details of the update rules of L-VSD. Concretely, L-VSD first updates $\phi_i$ with

$$\phi_{i+1} = \phi_i - 2\eta\Delta_{\phi_i}, \ \Delta_{\phi_i} := (\epsilon_{\phi_i}(x_{t'}, t', c, y) - \epsilon')J_{\phi_i}(x_{t'}, t', c, y) \tag{7}$$

where $\eta$ denotes the learning rate of the LoRA model.

Then, L-VSD updates $\theta_i$ with the updated LoRA model $\epsilon_{\phi_{i+1}}$:

$$\nabla_{\theta_i}\mathcal{L}_{L-VSD}(\theta_i) = \mathbb{E}_{t,\epsilon,c}\left[w(t)(\epsilon_{pretrain}(x_t, t, y) - \epsilon_{\phi_{i+1}}(x_t, t, c, y))\frac{\partial g(\theta, c)}{\partial\theta}\Big|_{\theta=\theta_i}\right]. \tag{8}$$

We can decompose $\epsilon_{\phi_{i+1}}(x_t, t, c, y)$ by Taylor series to understand the gap between the update rule of VSD (Equation (5)) and that of L-VSD (Equation (8)) for $\theta_i$:

$$\epsilon_{\phi_{i+1}}(x_t, t, c, y) = \epsilon_{\phi_i}(x_t, t, c, y) + (\phi_{i+1} - \phi_i)J^T_{\phi_i}(x_t, t, c, y) + \ldots$$
$$= \epsilon_{\phi_i}(x_t, t, c, y) + \underbrace{(-2\eta\Delta_{\phi_i}J^T_{\phi_i}(x_t, t, c, y))}_{\Delta\epsilon_{first}} + \underbrace{\mathcal{O}(\Delta^2_{\phi_i})}_{\Delta\epsilon_{high}}. \tag{9}$$

Namely, both the first-order term $\Delta\epsilon_{first}$ and the high-order one $\Delta\epsilon_{high}$ may contribute to the fast formation of geometry and texture in L-VSD.

To disentangle their effects, we opt to plot how their norms vary during the training procedure. In particular, we set $\eta$ to 0.01. We leave how to estimate $\Delta\epsilon_{first}$ to the next subsection and compute $\Delta\epsilon_{high}$ by $\epsilon_{\phi_{i+1}}(x_t, t, c, y) - \epsilon_{\phi_i}(x_t, t, c, y) - \Delta\epsilon_{first}$. As illustrated in Fig. 6, $||\Delta\epsilon_{high}||$ is significantly larger than $||\Delta\epsilon_{first}||$. It is even larger than the norm of the entire LoRA model $||\epsilon_\phi||$, which means that $\epsilon_{\phi_{i+1}}$ is probably dominated by $\Delta\epsilon_{high}$. Moreover, the norm of $\Delta\epsilon_{high}$ varies in a considerable range, which indicates much greater randomness than the first-order term.

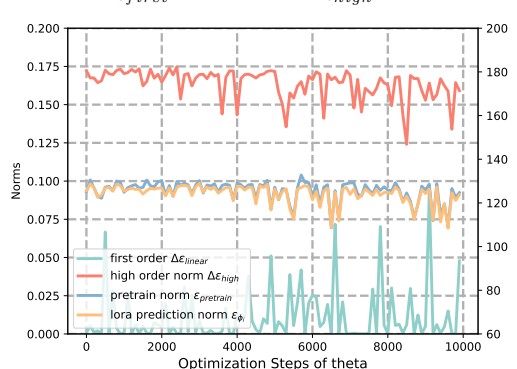

Figure 6: **The norm of various scores during the training of L-VSD.**

To obtain a more intuitive understanding of the two terms, we also pass them through the decoder of the pretrained DM (because we experiment with latent DMs (Rombach et al., 2022)) to obtain visualizations of them, presented in Fig. 5. As shown, the visualization of $\Delta\epsilon_{first}$ indicates an object shape corresponding to the prompt while $\Delta\epsilon_{high}$ is more random and we cannot witness any semantic information within it.

Based on all the above observations and inferences, we wonder **whether $\Delta\epsilon_{first}$ is the essential component for the appealing generations, and keeping it can improve score distillation further.** We answer to this in the next subsection.

## 4.2 LINEARIZED LOOKAHEAD VSD

Let $\epsilon_{\phi_{i+1}}^{\lin}(x_t, t, c, y) := \epsilon_{\phi_i}(x_t, t, c, y) + \Delta\epsilon_{first}$. In fact, it corresponds to performing one-step SGD with $(x_{t'}, t')$ under the denoising loss (Equation (2)) to update the following linear noise-prediction model:

$$\epsilon_\phi^{\lin}(x_t, t, c, y) = \epsilon_{\phi_i}(x_t, t, c, y) + (\phi - \phi_i)J_{\phi_i}^T(x_t, t, c, y). \tag{10}$$

Due to the low complexity of the linear model, the risk of overfitting to the current training data is low, which properly addresses the problem of the original L-VSD. With these understandings, our method boils down to using only the linearized lookahead correction $\epsilon_{\phi_{i+1}}^{\lin}(x_t, t, c, y)$ for estimating the score function of noisy rendered images of $\theta_i$. i.e.,

$$\nabla_{\theta_i}\mathcal{L}^*(\theta_i) = \mathbb{E}_{t,\epsilon,c}\left[w(t)\left(\epsilon_{pretrain}(x_t, t, y) - \epsilon_{\phi_{i+1}}^{\lin}(x_t, t, c, y)\right)\frac{\partial g(\theta, c)}{\partial\theta}\Big|_{\theta=\theta_i}\right]$$

$$= \mathbb{E}_{t,\epsilon,c}\left[w(t)\left(\epsilon_{pretrain}(x_t, t, y) - \epsilon_{\phi_i}(x_t, t, c, y) + 2\eta\Delta_{\phi_i}J_{\phi_i}^T(x_t, t, c, y)\right)\frac{\partial g(\theta, c)}{\partial\theta}\Big|_{\theta=\theta_i}\right]$$

$$= \mathbb{E}_{t,\epsilon,c}\Big[w(t)\Big(\epsilon_{pretrain}(x_t, t, y) - \epsilon_{\phi_i}(x_t, t, c, y)$$
$$+ 2\eta(\epsilon_{\phi_i}(x_{t'}, t', c, y) - \epsilon')\left[J_{\phi_i}(x_{t'}, t', c, y)J_{\phi_i}^T(x_t, t, c, y)\right]\Big)\frac{\partial g(\theta, c)}{\partial\theta}\Big|_{\theta=\theta_i}\Big]. \tag{11}$$

Viewing $J_{\phi_i}(x_{t'}, t', c, y)J_{\phi_i}^T(x_t, t, c, y) \in \mathbb{R}^{d\times d}$ as a pre-conditioning matrix, the above update rule involves two score contrast terms—one corresponds to the original VSD objective and the other accounts for a linearized lookahead correction for practical iterative optimization.

Moreover, we clarify the term $\Delta\epsilon_{first} := -2\eta\Delta_{\phi_i}J_{\phi_i}^T(x_t, t, c, y)$ is friendly to estimate. Concretely, $\Delta_{\phi_i}$ is exactly the gradient to update the LoRA model, which a backward pass can calculate. Then, the vector-Jacobian product $\Delta_{\phi_i}J_{\phi_i}^T(x_t, t, c, y)$ can be realized by a forward pass of the score model using forward-mode automatic differentiation, which is equipped in many deep learning frameworks (Paszke et al., 2017) As a result, $L^2$-VSD only needs a single additional forward pass of the LoRA model per iteration compared to VSD, which is much more efficient than updating the LoRA multiple times. More importantly, we can even use only the entities associated with the last layer of the LoRA model for estimating the vector-Jacobian product to achieve a trade-off between quality and efficiency.

## 5 EXPERIMENTS

### 5.1 SETTINGS

In this section, we evaluate the efficacy of our proposed Linearized Lookahead VSD method on text-guided 3D generation. Our baseline approaches include SDS, VSD, ESD (Entropy Score Distillation) (Wang et al., 2024a), and VSD-based HiFA (Zhu et al., 2023), which include representative state-of-the-art methods. For a fair comparison, all experiments are benchmarked under the open-source framework threestudio (Guo et al., 2023). To clearly demonstrate the superior performance brought by our method, we compare both the results produced with high resolution and low resolution. Please refer to Appendix B.1 for more implementation details.

### 5.2 QUALITATIVE COMPARISON

We compare our generation results with SDS, VSD, ESD, L-VSD and HiFA both in high resolution of 256 and low resolution of 64. We present some representative results without bias in Fig. 7 and

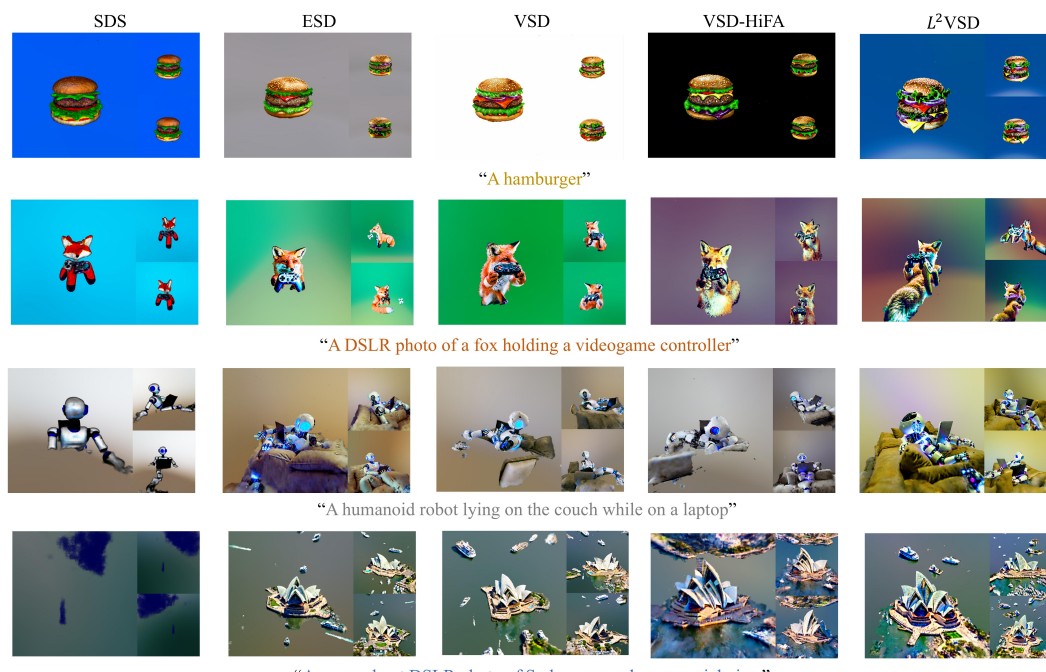

Figure 7: **Qualitative comparison with high resolution of 256**. $L^2$-VSD can generate meticulously detailed and realistic 3D assets from easy to very complex prompts and scenes. $L^2$-VSD tends to yield more complete structures than HiFA.

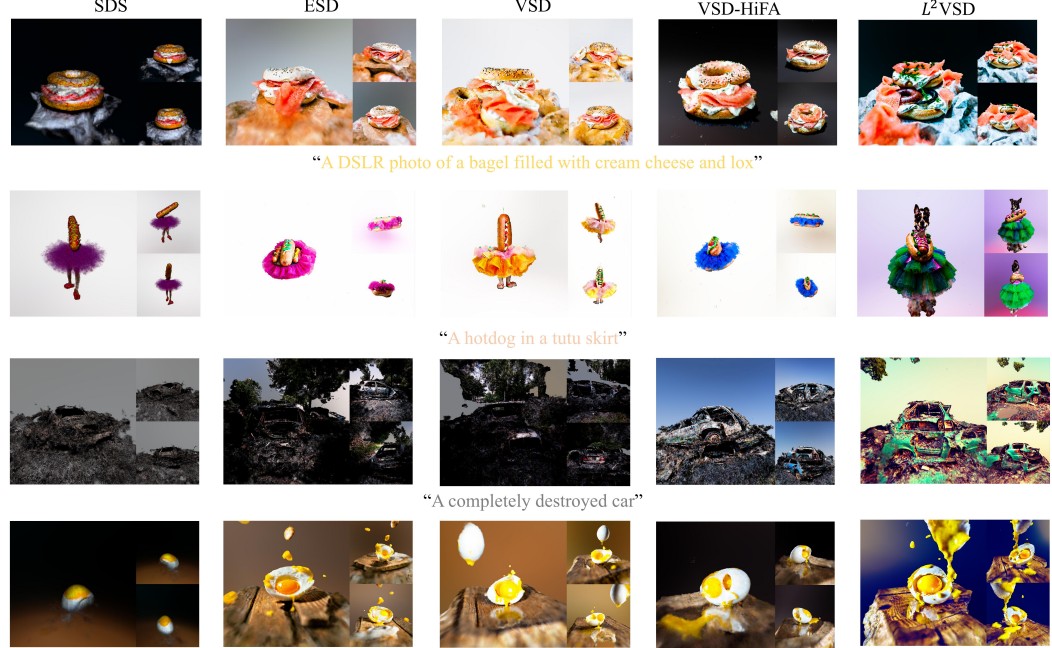

Figure 8: **More Qualitative comparison with high resolution**.

Fig. 15, and we refer readers to Appendix C for results of L-VSD and more cases, as L-VSD tends to fail in generating complete objects. It's noteworthy that we consider our method as a baseline similar to SDS and VSD. ESD and HiFA are state-of-the-arts based on VSD, which try to improve certain properties of generation quality by introducing novel techniques. We compare with them in order to position our performance more rigorously. We demonstrate in Sec. 5.5 that our methods can be seamlessly combined with other techniques like ESD and HiFA. Compared with VSD and its variant ESD, our results have significantly more realistic appearances and reasonable geometry. Moreover, our method has better convergence, which means our results are more robust to optimization and do

not suffer from generating floaters in space. It's clearly shown that with our linearized lookahead correction term, even only trained with low-resolution rendering settings, we can generate 3D scenes with realistic and detailed appearances. In conclusion, our method surpasses the other baseline methods significantly with linearized lookahead and achieves comparable results with state-of-the-art.

## 5.3 QUANTITATIVE COMPARISON

We follow previous work (Poole et al., 2022; Yu et al., 2023) to quantitatively evaluate the generation quality using the angle of CLIP similarity (Hessel et al., 2021) and Frechet Inception Distance (Heusel et al., 2017) and list the results in Table 1. Specifically, the CLIP similarity measures the cosine similarity between the rendered image embeddings of the generated 3D object and the text embeddings, and we calculate the angle. The FID measures the distance between the image distribution by randomly rendering 3D representation and the text-conditioned image distribution from the pretrained diffusion model. We use prompts from the gallery of DreamFusion to ensure no man-made bias in evaluation. Please refer to Appendix B.1 for more metric computation details.

Table 1: **Quantitative Comparison.** ($\downarrow$) means the lower the better. We measure the scenes across 20 prompts, which are randomly sampled from DreamFusion's gallery and include scenes shown in Fig. 7, Fig. 15 and Fig. 8. The quantitative results align with the qualitative results.

|  | SDS | ESD | VSD | L-VSD | HiFA | $L^2$-VSD |
|---|---|---|---|---|---|---|
| Averaged CLIP sim ($\downarrow$) | 0.305 | 0.316 | 0.324 | 0.337 | 0.313 | **0.285** |
| Averaged FID ($\downarrow$) | 372.35 | 315.15 | 301.54 | 496 | 292.88 | **284.06** |

## 5.4 ABLATION STUDY

### 5.4.1 ABLATION ON $\eta$

We conduct an ablation study on the choice of $\eta$, showing the results in Fig. 9a, where only $\eta$ changes while other parameters are unchanged. We still use the simple prompt "a delicious hamburger". It's demonstrated that our method is robust to the scale of $\eta$. It's worth noting that even if the norm of first-order term $\Delta\epsilon_{first}$ is only at the scale of 1e-2 when we set $\eta$ to 1e-3, the results still improve a lot. So, even a minor correction for each iteration can lead to incredible improvement in such a long-term optimization. Ablation on high-order term correction can be found in Appendix C.1.

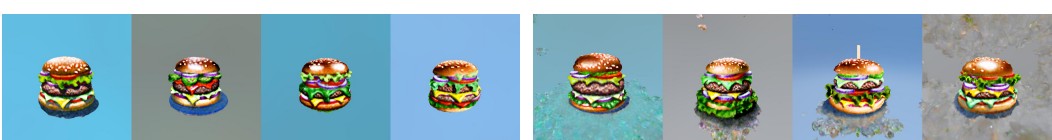

(a) **Ablation on the $\eta$ in $\Delta\epsilon_{first}$.** From left to right, the results correspond to setting $\eta$ to 1e-3, 1e-2, 0.1, and 1 respectively.

(b) **Ablation on last-layer approximation.** From left to right, the results also correspond to those in (a), but use a last-layer approximation for estimating $\Delta\epsilon_{first}$.

Figure 9: **Qualitative results of ablation studies.**

### 5.4.2 ABLATION ON LAST-LAYER APPROXIMATION

As stated above, we can use only the entities associated with the last layer of the LoRA model to further reduce additional time costs. Admittedly, this approach may result in performance loss. As the correction term is calculated with the Jacobian matrixs of LoRA model, if only use the last-layer gradients to approximate the correction, we actually ignore the variable updates within other layers, thus losing accuracy. We present the corresponding results in Fig. 9b. Although the outcomes tend to generate floaters, the realism of results still gets largely improved. Also, we believe the floaters can be eliminated in the following geometry refinement stage. We compare the computation efficiency with the baseline VSD in Table 2. We can save much computation time cost by only using this approximation.

Table 2: **Computation efficiency.** We present the time cost in each iteration. We measure the average time on the threestudio framework.

|  | Time cost (s/iteration) |
|---|---|
| VSD | $\sim 0.7$ |
| $L^2$-VSD | $\sim 1.0$ |
| $L^2$-VSD (last-layer) | $\sim 0.8$ |

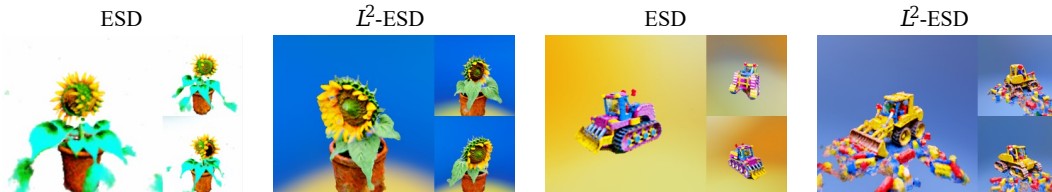

Figure 10: **Examples of combining ESD with $L^2$-VSD.** We can observe that in the case of "sunflower", by combining our method, we obtain a reasonable sunflower rather than a "sunball". In the case of "bulldozer", with a linearized lookahead, ESD can generate additional elements like bricks.

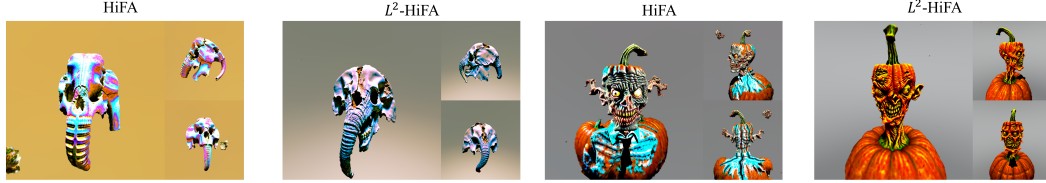

Figure 11: **Examples of combining HiFA with $L^2$-VSD.** The colors are more realistic and the appearance of our method's results aligns better with the prompts.

## 5.5 CONNECTION WITH OTHER DIFFUSION DISTILLATION METHODS

**ESD** (Wang et al., 2024a) is dedicated to solving the Janus problem based on VSD. It maximizes the entropy of different views of the generated results, encouraging diversity across views. The gradient of ESD is theoretically equivalent to adopting CFG (Ho & Salimans, 2022) trick upon VSD, i.e.

$$\nabla_{\theta_i} \mathcal{L}_{ESD}(\theta_i) = \mathbb{E}_{t,\epsilon,c} \left[ w(t)(\epsilon_{pretrain}(x_t,t,y) - \lambda \epsilon_{\phi_i}(x_t,t,\emptyset,y) - (1-\lambda)\epsilon_{\phi_i}(x_t,t,c,y)) \frac{\partial g(\theta,c)}{\partial \theta}\Big|_{\theta=\theta_i} \right],$$

(12)

where $\lambda$ is a scaling factor. Our method can also be incorporated into ESD. We name this as $L^2$-ESD. We present the corresponding results in Fig. 10. The prompts we use are "a sunflower on a flowerpot" and "a bulldozer made out of toy bricks" respectively. To avoid unnecessary trials, we simply set $\lambda$ to 0.5, which is recommended in ESD. As we can observe, compared with ESD, the 3D results are more photo-realistic with distinct shapes.

**HiFA** (Zhu et al., 2023) is a state-of-the-art approach for high-quality 3D generation in a single-stage training based on basic methods. It distills denoising scores from pretrained models in both the image and latent spaces and proposes several techniques to improve NeRF generation, which are orthogonal to our method. We demonstrate our method's compatibility by combining with HiFA, naming as $L^2$-HiFA. We present the results in Fig. 11. The prompts we use are "an elephant skull" and "Pumpkin head zombie, skinny, highly detailed, photorealistic" respectively.

In conclusion, it's believed that $L^2$-VSD can be combined with other parallel techniques in the future.

## 6 DISCUSSION, CONCLUSION AND LIMITATION

In this paper, we dive deep into the theory of VSD, having a comprehensive understanding of inner process and identify two potential directions for improvement. In terms of the algorithm formulation, we propose Linearized Lookahead Variational Score Distillation($L^2$-VSD), a novel framework based on VSD that achieves state-of-the-art results on text-to-3D generation. The linearized lookahead term enables us to benefit from both better convergence and lookahead for next iteration. More importantly, our method can be incorporated into any VSD-based framework in the future.

**Limitations and broader impact.** Firstly, although $L^2$-VSD achieves remarkable improvement on text-to-3D results, the generation process still takes hours of time, which is a common issue for score distillation based methods. We believe orthogonal improvements on distillation accelerating (Zhou et al., 2023) can mitigate this problem. Secondly, while we find the first-order term shows regular pattern and some similarity may exist between our method and SiD (Zhou et al., 2024), which is briefly discussed in Appendix E, we have not yet been able to formulate a distribution-based objective that guides this optimization. Investigating the underlying reasons is of significant interest in the future. Lastly, as for broader impact, like other generative models, our method may be utilized to generate fake and malicious contents, which needs more attention and caution.

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

# A    ILLUSTRATIVE OVERVIEW

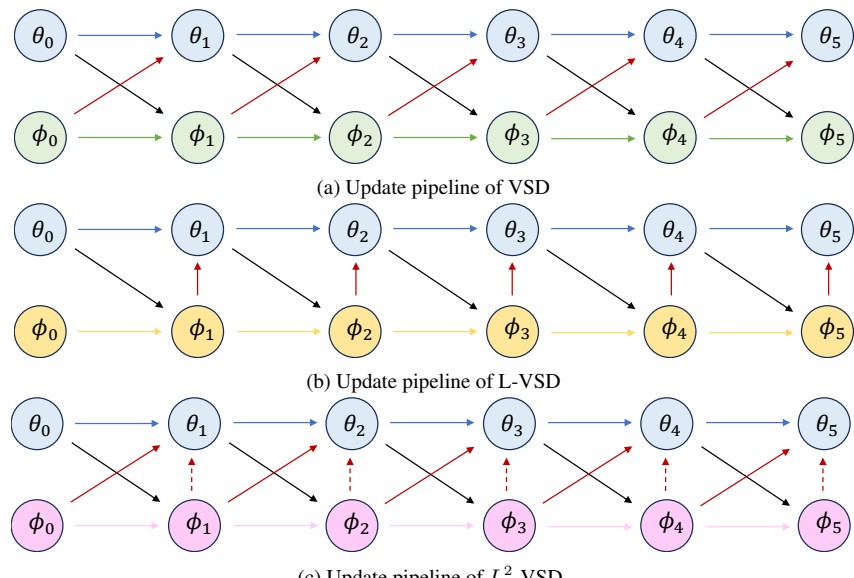

(a) Update pipeline of VSD

(b) Update pipeline of L-VSD

(c) Update pipeline of $L^2$-VSD

Figure 12: **Overview of VSD, L-VSD and $L^2$-VSD training.**

We present an illustrative overview about the updating pipeline of VSD, L-VSD and $L^2$-VSD respectively. As stated in Sec 2.2, we use $\theta_i$ and $\phi_i$ to represent the 3D and the LoRA models at $i_{th}$ iteration respectively. We use arrows with different colors to represent state transition dependency. We argue that red dashed arrow pointing from $\phi_i$ to $\theta_i$ is important for better results' quality.

**More illustrative 2D gaussian examples.** To gain a more complete view about the convergence of VSD, we conduct two additional gaussian experiments as shown in Fig. 13 and Fig. 14. In the example of Sec. 3.2, we only sample one point to keep as the same in ProlificDreamer, in which only one view of 3D object is rendered. In Fig. 13, we increase the number to 4, finding that the error introduced by optimization order could be mitigated to some extent. This evidence enlightens us that VSD with multi-view estimation may perform better, part of which has been proved in MVDream (Shi et al., 2023). Besides, we also show the bad convergence if we overfit LoRA model on current sampled views in Fig. 14. It's worth noting that the distribution tends to lie between the intersection of two gaussian modals, making the views more saturated, which is coherent to the finding in Sec. 3.3. We provide the reproducible example code in Appendix. F.

# B    EXPERIMENT IMPLEMENTATION

## B.1    MAIN EXPERIMENTS DETAILS

**Qualitative Results.** In this section, we provide more details on the implementation of $L^2$-VSD and the compared baseline methods. All of them are implemented under the threestudio framework directly in the first stage coarse generation, without geometry refinement and texture refinement, following (Wang et al., 2024b). For the coarse generation stage, we adopt foreground-background disentangled hash-encoded NeRF (Müller et al., 2022) as the underlying 3D representation. All scenes are trained for 15k steps for the coarse stage, in case of geometry or texture collapse. At each interation, we randomly render one view. Different from classic settings, we adjust the rendering resolution directly as $64 \times 64$ in the low resolution experiments. And increase to $256 \times 256$ resolution in the high resolution experiments. All of our experiments are conducted on a single NVIDIA GeForce RTX 3090.

**Quantitative Results.** To compute FID (Heusel et al., 2017), we sample $N$ images using pretrained latent diffusion model given text prompts as the ground truth image dataset, and render $N$ views uniformly distributed over a unit sphere from the optimized 3D scene as the generated image dataset. Then standard FID is computed between these two sets of images. To compute CLIP similarity, we

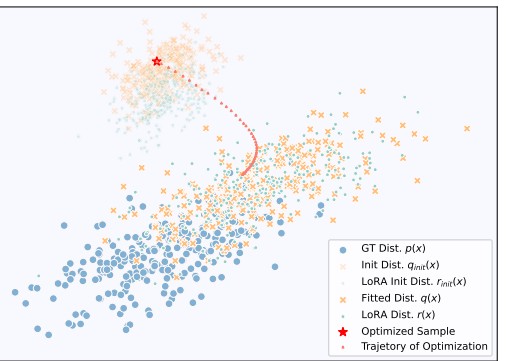 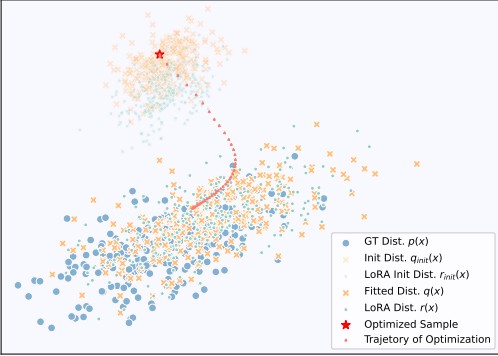

Figure 13: **Comparison of VSD and L-VSD with more render samples.** In this example, we sample 4 points in each iteration.

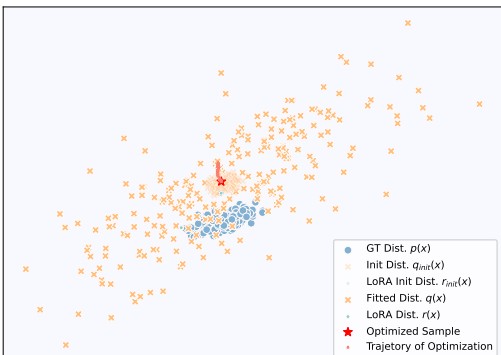 , 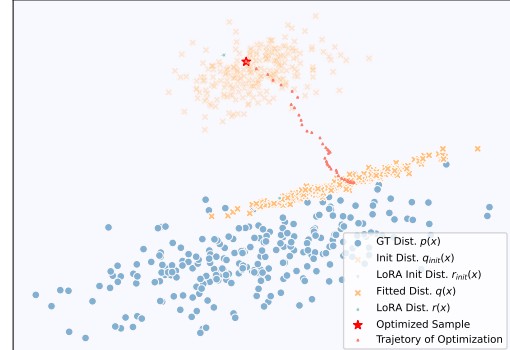

Figure 14: **Exploring the impact of $r(x)$ overfitting on rendered samples.** In this example, $r(x)$ is delta distribution as we overfit it on $x$ in each iteration.

render 120 views from the generated 3D representations, and for each view, we obtain an embedding vector and text embedding vector through the image and text encoder of a CLIP model. We use the CLIP ViT-B/16 model (Radford et al., 2021).

## B.2 HIGH ORDER $\Delta\epsilon_{high}$ OMPUTATION DETAILS

As mentioned above in Sec.4.1, we can comppute $\Delta\epsilon_{high}$ as $\epsilon_{\phi_{i+1}}(x_t, t, c, y) - \epsilon_{\phi_i}(x_t, t, c, y) - \Delta\epsilon_{first}$. In practice, we implement this computation during the training process of L-VSD. We copy an additional LoRA model to restore the LoRA parameters before being updated. Then in each optimization iteration for $\theta_i$, the LoRA model performs forward passes for three times to calculate the $\epsilon_{\phi_i}$, $\epsilon_{\phi_{i+1}}$ and $\Delta\epsilon_{first}$ respectively.

## C MORE EXPERIMENT RESULTS

### C.1 FAILURE CASES PRODUCED BY L-VSD

We show an example of failure case produced by L-VSD in Fig. 16. We can observe that the upper one becomes over-saturated faster than the below one. Though the below one collapses much slower, it can't converge to a realistic case. Also, we provide all the L-VSD results in Fig. 17, which reflects the unstable generation quality by naive L-VSD.

### C.2 GENERALIZATION ON OTHER REPRESENTATIONS

We provide the results generated in the second "geometry refinement" and third "texture refinement" stage in Fig. 18 and Fig. 19. In Fig. 18, the 3D objects are initialized with the results in the first stage. While in Fig. 19, we control the geometry initialization to be the same for our method and VSD, thus directly comparing the texture generation quality. In Fig. 19, VSD generates destroyed car with random red color, connecting destroyed car with a fire but our method generates more purely. And

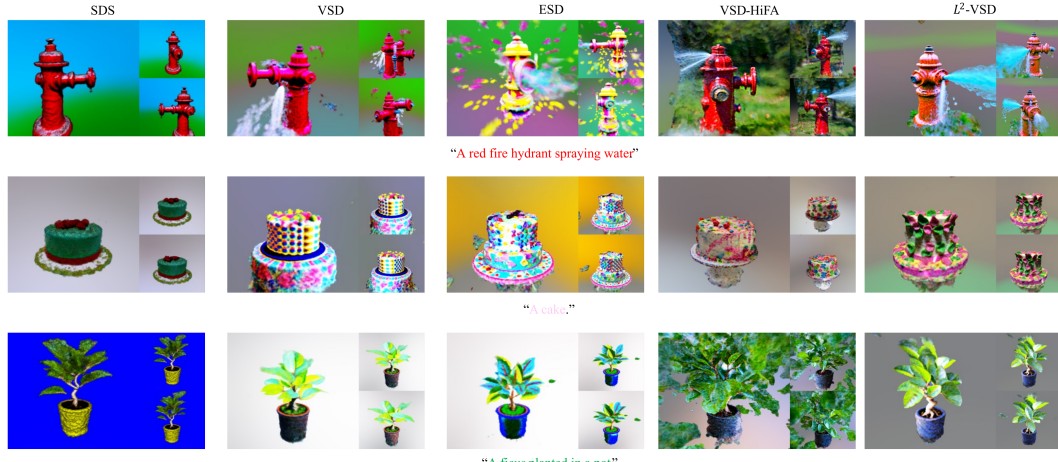

Figure 15: **Qualitative comparison with low resolution of 64**. $L^2$-VSD can generate highly detailed 3D assets even with low resolution, while the other baselines (except for HiFA), suffering from geometry-texture co-training, tend to be blurry and have floaters.

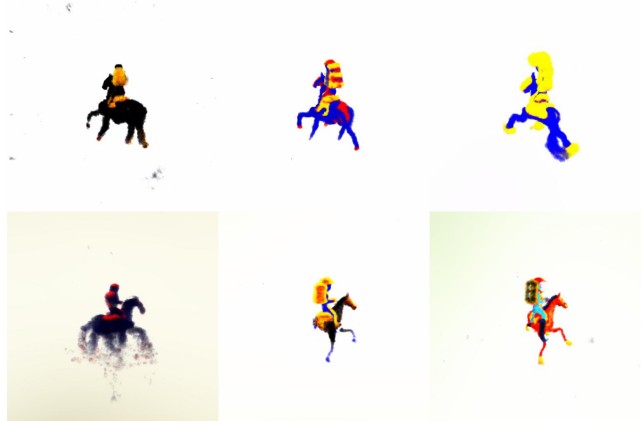

Figure 16: **Visualization of Failure Process.** The upper row result is generated with original learning rate while the lower one is generated with scaling the learning rate by 0.1. Each row corresponds to a continue optimization process. Our prompt is "an astronaut riding a horse".

the texture of hand and the bowl in the bottom is also more realistic. As these two stages represent in mesh, we believe this comparison reflects the generalization of our method on other representations.

### C.3 LOSS CURVE COMPARISON AT INITIAL STAGES

As requested by Reviewer YuaJ, we show the loss curve in Fig. 20a. As shown by the curve, the loss is in similar level at the start of distillation, which is probably because the objects don't form into clear shape yet. So the predicted noises are all likely to be gaussian.

Also, as suggested by Review LcCM, we test on multiple samples and measure the average LoRA loss to provide more convincing results, which is shown in Fig. 20b. The conclusion holds as the same as in the section. 3.1. Also, we provide one sample "crown" other than "hamburger" to augment the proof.

### C.4 ABLATION OF GENERATION WITH HIGH-ORDER TERM

We provide the results of one important ablation experiment in Fig. 22. We compare the results produced by VSD, $L^2$-VSD and HL-VSD(high-order lookahead VSD). In HL-VSD, we use the

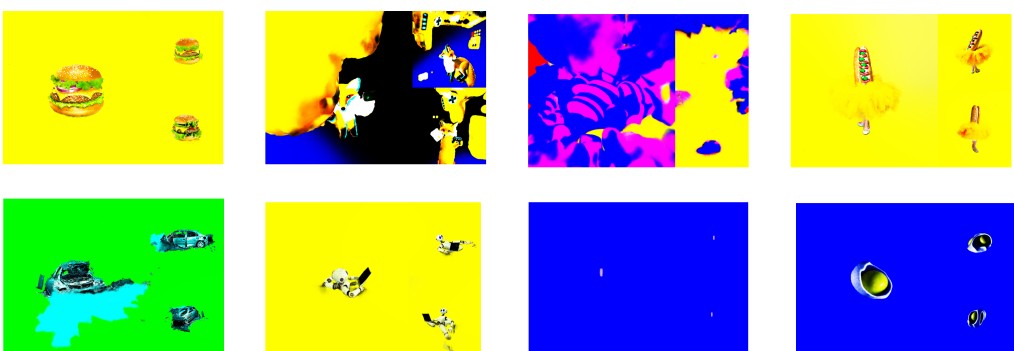

Figure 17: **Results of L-VSD.** These results are generated with the same prompts in Fig. 7 and Fig. 15. As we can observe, naive L-VSD usually fails in generating realistic objects, which is supported by our Gaussian example in Sec. 3.2.

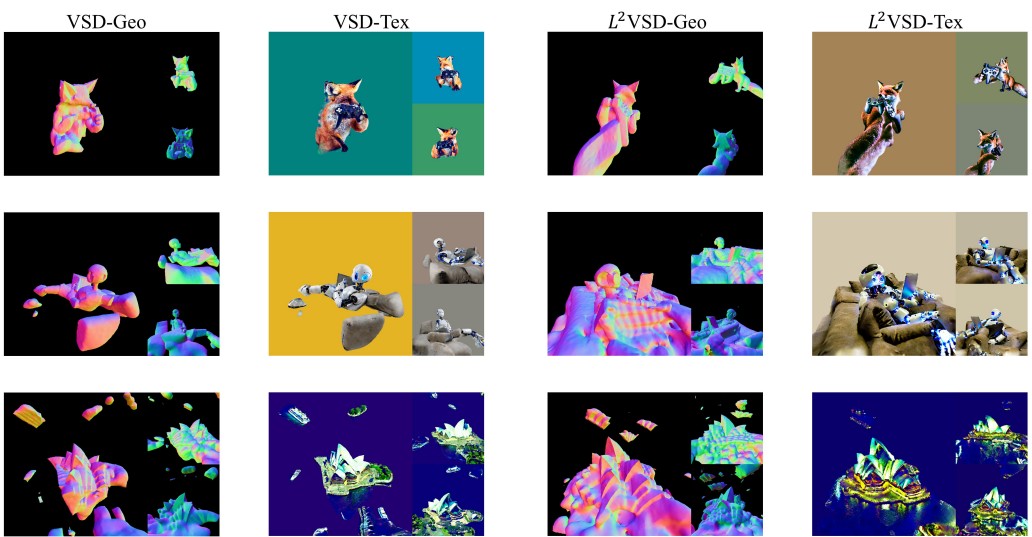

Figure 18: **Comparison at second and third stages**. We initial the objects with first-stage's results and compare the geometry and texture refinement. As shown in the figure, the geometry generated by our method is more complete and texture generated by our method is much more realistic.

high-order term instead of the linear term to correct the score. As shown in the figure, the results all collapse and become irrecognizable, which proves the effectiveness and necessity of linearied lookahead.

# D    OTHER RELATED WORKS

## D.1    TEXT-TO-IMAGE DIFFUSION MODELS

Text-to-image diffusion models (Ramesh et al., 2021; 2022) are essential for text-to-3D generation. These models incorporate text embeddings during the iterative denoising process. Leveraging large-scale image-text paired datasets, they address text-to-image generation tasks. Latent diffusion models (Rombach et al., 2022), which diffuse in low-resolution latent spaces, have gained popularity due to reduced computation costs. Additionally, text-to-image diffusion models find applications in various computer vision tasks, including text-to-3D (Ramesh et al., 2022; Singer et al., 2023), image-to-3D (Xu et al., 2023a), text-to-svg (Jain et al., 2023), and text-to-video (Khachatryan et al., 2023; Singer et al., 2022).

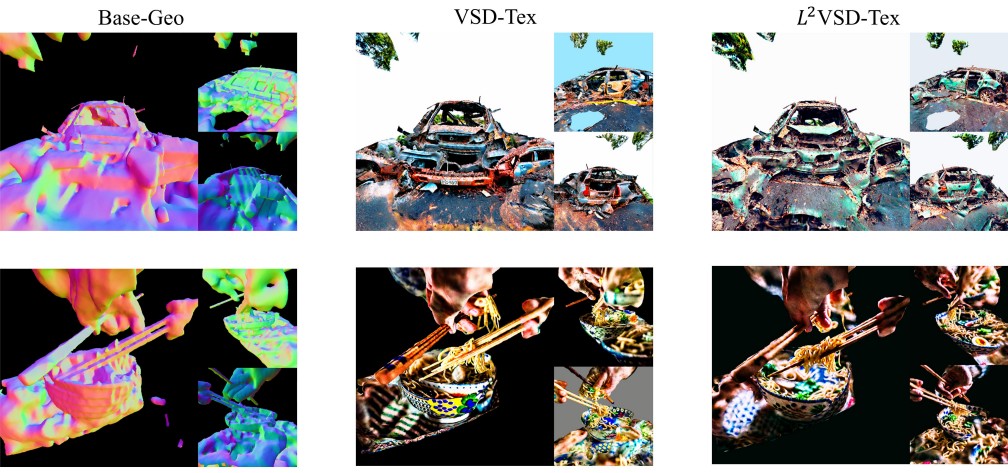

Figure 19: **Comparison on texture representation**. We use VSD and our method to generate texture conditioned on the same geometry initialization. Prompts: (Upper)"a completely destroyed car" ;(Bottom)"a zoomed out DSLR photo of a pair of floating chopsticks picking up noodles out of a bowl of ramen".

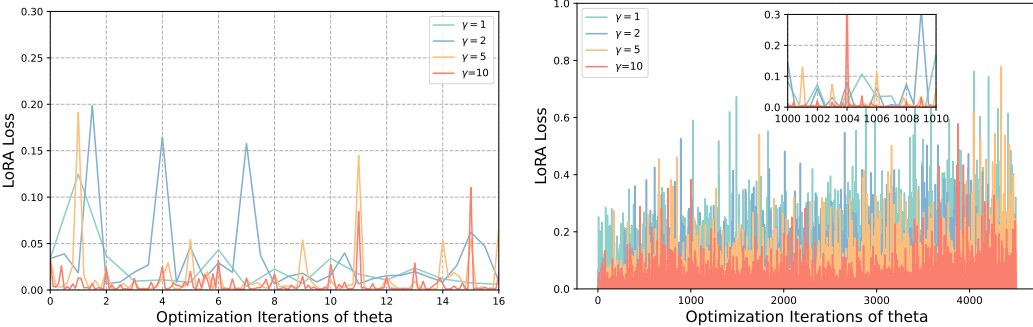

(a) **VSD multi-LoRA initial loss**: At the start of distillation, the loss with different LoRA steps is in the similar level.

(b) **Multi samples averaged loss curve.** We average the LoRA loss on 3 samples, finding the general pattern of loss variation.

Figure 20: **More Loss Curve.**

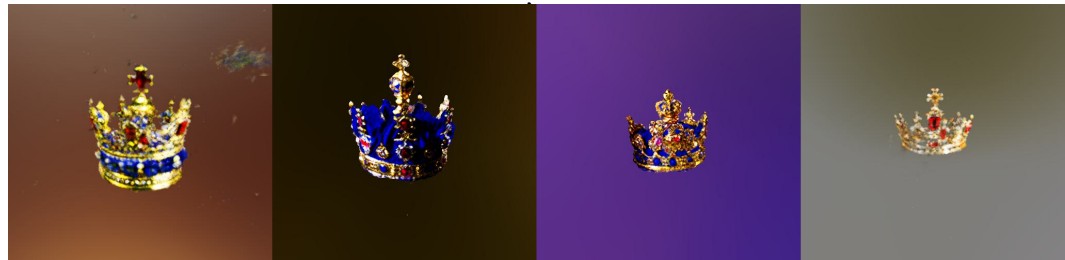

Figure 21: **VSD LoRA Comparison**.

## D.2 TEXT-TO-3D GENERATION WITHOUT 2D-SUPERVISION

Text-to-3D generation techniques have evolved beyond relying solely on 2D supervision. Researchers explore diverse approaches to directly create 3D shapes from textual descriptions. Volumetric representations, such as 3D-GAN (Sun et al., 2020) and Occupancy Networks (Mescheder et al., 2019), use voxel grids (Sun et al., 2022; Liu et al., 2019). Point cloud generation methods, like PointFlow (Yang et al., 2019) and AtlasNet (Vakalopoulou et al., 2018), work with sets of 3D points. Implicit surface representations, exemplified by DeepVoxels (Sitzmann et al., 2019) and SIREN (Sitzmann et al., 2020), learn implicit functions for shape surfaces. Additionally, graph-based

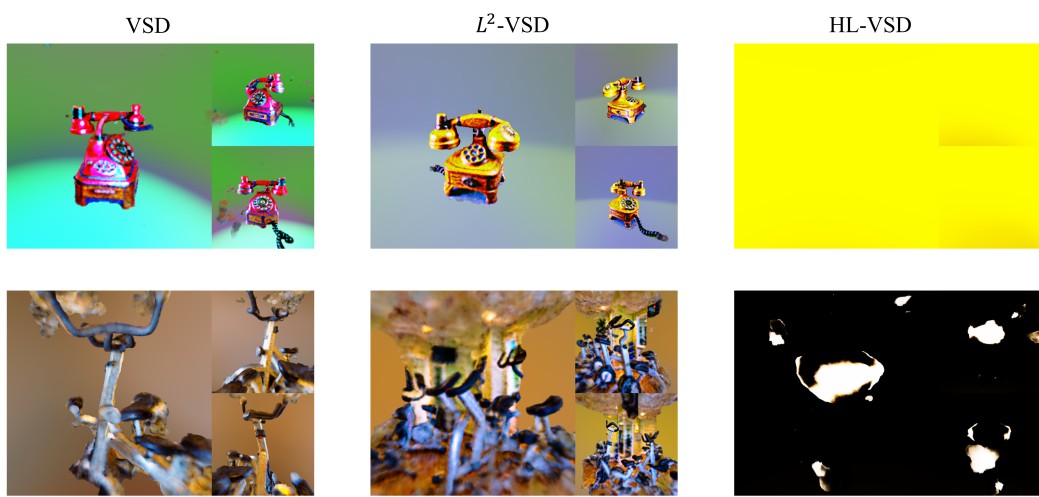

Figure 22: **Results comparison with using high-order term**. Prompts: (upper)"A rotary telephone carved out of wood" ;(Bottom)"a DSLR photo of an exercise bike in a well lit room"

approaches (GraphVAE (Simonovsky & Komodakis, 2018), GraphRNN (You et al., 2018)) capture relationships between parts using graph neural networks.

### D.3  ADVANCEMENTS IN 3D SCORE DISTILLATION TECHNIQUES

Various techniques enhance score distillation effectiveness. Magic3D (Lin et al., 2023) and Fantasia3D (Chen et al., 2023) disentangle geometry and texture optimization using mesh and DMTet (Shen et al., 2021). TextMesh (Tsalicoglou et al., 2023) and 3DFuse (Seo et al., 2023) employ depth-conditioned text-to-image diffusion priors for geometry-aware texturing. Score debiasing (Hong et al., 2023) and Perp-Neg (Zhao et al., 2023) refine text prompts for better 3D generation. Researchers also explore timestep scheduling (DreamTime (Huang et al., 2023), RED-Diff (Mardani et al., 2023)) and auxiliary losses (CLIP loss (Xu et al., 2023b), adversarial loss (Oikarinen et al., 2021)) to improve score distillation.

### E  DISCUSSION

**Score Identity Distillation (SiD) (Zhou et al., 2024)** Apart from direct comparison with the text-to-3D score distillation method, our method can draw some similarities with some 2D diffusion distillation methods. SiD reformulates forward diffusion as semi-implicit distributions and leverages three score-related identities to create an innovative loss mechanism. The weighted loss is expressed as:

$$\tilde{\mathcal{L}}_{SiD}(\theta_i) = -\alpha \frac{w(t)}{\sigma_t^4} ||\epsilon_{pretrain}(x_t, t) - \epsilon_\phi(x_t, t)||_2^2$$
$$+ \frac{w(t)}{\sigma_t^4} (\epsilon_{pretrain}(x_t, t) - \epsilon_\phi(x_t, t))^T (\epsilon_\phi(x_t, t) - \epsilon) \tag{13}$$

where $x_t = g(\theta_i)$. Compared with the original VSD loss, the additional term in SiD has an important factor $(\epsilon_\phi - \epsilon)$, which corrects the original loss in a projected direction. This factor also exists in our term, so we assume that our first-order term shares some similarity with this correction term.

### F  GAUSSIAN EXAMPLE CODE

```
1 import os
2 import math
3 import random
4 import numpy as np
```

```python
from tqdm import tqdm, trange
import matplotlib.pyplot as plt

import torch
import torch.nn as nn
import torch.nn.functional as F
from torch.optim.lr_scheduler import LambdaLR

def get_cosine_schedule_with_warmup(optimizer, num_warmup_steps,
        num_training_steps, min_lr=0., num_cycles: float = 0.5):

    def lr_lambda(current_step):
        if current_step < num_warmup_steps:
            return float(current_step) / float(max(1, num_warmup_steps))
        progress = float(current_step - num_warmup_steps) / float(max(1,
            num_training_steps - num_warmup_steps))
        return max(min_lr, 0.5 * (1.0 + math.cos(math.pi *
            float(num_cycles) * 2.0 * progress)))

    return LambdaLR(optimizer, lr_lambda, -1)

def seed_everything(seed):
    random.seed(seed)
    os.environ['PYTHONHASHSEED'] = str(seed)
    np.random.seed(seed)
    torch.manual_seed(seed)
    torch.cuda.manual_seed(seed)

def sample_gassian(mu, sigma, N_samples=None, seed=None):
    assert N_samples is not None or seed is not None
    if seed is None:
        seed = torch.randn((N_samples, d), device=mu.device)
    samples = mu + torch.matmul(seed, sigma.t())
    return samples

# Core function: compute score function of perturbed Gaussian
    distribution
# \nabla \log p_t(x_t) = -(Simga^{-1} + sigma_t^2 I) (x_t - \alpha_t *
    \mu)
def calc_perturbed_gaussian_score(x, mu, sigma, alpha_noise,
        sigma_noise):
    if mu.ndim == 1:
        mu = mu[None, ...] # [d] -> [1, d]
    if sigma.ndim == 2:
        sigma = sigma[None, ...] # [d, d] -> [1, d, d]

    mu = mu * alpha_noise[..., None] # [B, d]
    sigma = torch.matmul(sigma, sigma.permute(0, 2, 1)) # [1, d, d]
    sigma = (alpha_noise**2)[..., None, None] * sigma # [B, d, d]
    sigma = sigma + (sigma_noise**2)[..., None, None] *
        torch.eye(sigma.shape[1], device=sigma.device)[None, ...] # [B,
        d, d]
    inv_sigma = torch.inverse(sigma) # [B, d, d]
    return torch.matmul(inv_sigma, (mu - x)[..., None]).squeeze(-1) # [B,
        d, d] @ [B, d, 1] -> [B, d, 1] -> [B, d]

# data dimension
N = 256
d = 2
ndim = d
lora_steps = 10
# set the hyperparameters
seed = 0
dist_0 = 10
lr = 1e-2
```

```
61  min_lr = 0
62  weight_decay = 0
63  warmup_steps = 100
64  total_steps = 2000
65  scheduler_type = 'cosine'
66  lambda_coeff = 1.0
67  method = 'l-vsd' # or 'real-vsd', 'vsd'
68  output_dir = ''
69  logging_steps = 10
70
71  device = torch.device('cuda:0')
72  seed_everything(seed)
73
74  # groundtruth distribution
75  p_mu = torch.rand(d, device=device) # uniform random in [0, 1] x [0, 1]
76  p_sigma = torch.rand((d, d), device=device) + torch.eye(d,
        device=device) # positive semi-definite
77
78  # diffusion coefficients
79  beta_start = 0.0001
80  beta_end = 0.02
81
82  # parametric distribution to optimize
83  q_mu = nn.Parameter(torch.rand(d, device=device) * dist_0 + p_mu)
84  q_sigma = nn.Parameter(torch.rand(d, d, device=device))
85
86  r_mu = nn.Parameter(torch.zeros(d, device=device)).to(device)
87  r_sigma = nn.Parameter(torch.zeros(d, d, device=device)).to(device)
88
89  optimizer = torch.optim.AdamW([q_mu, q_sigma], lr=lr,
        weight_decay=weight_decay)
90  scheduler = get_cosine_schedule_with_warmup(optimizer, warmup_steps,
        int(total_steps*1.5), min_lr) if scheduler_type == 'cosine' else None
91
92  # set the optimizer and scheduler of LoRA model
93  r_optimizer = torch.optim.AdamW([r_mu, r_sigma], lr=5*lr,
        weight_decay=weight_decay)
94
95  # saving checkpoints
96  state_dict = []
97  N_render = 4
98  # store per-step samples. fixed seed for visualization
99  vis_seed = torch.randn((1, N, d), device=device)
100 vis_seed_true = torch.randn((1, N, d), device=device)
101 vis_seed2 = torch.randn((1, N, d), device=device)
102 vis_samples = [] # [steps, p+q, N_samples, N_dim]
103 # x_previous = 0
104
105 for i in trange(total_steps + 1):
106     optimizer.zero_grad()
107
108     # sample time steps and compute noise coefficients
109     betas_noise = torch.rand(N_render, device=device) * (beta_end -
            beta_start) + beta_start
110     alphas_noise = torch.cumprod(1.0 - betas_noise, dim=0)
111     sigmas_noise = ((1 - alphas_noise) / alphas_noise) ** 0.5
112
113     # sample from g(x) = q_mu + q_sigma @ c, c ~ N(0, I)
114     x = sample_gassian(q_mu, q_sigma, N_samples=N_render)
115     # sample gaussian noise
116     eps = torch.randn((N_render, d), device=device)
117     # diffuse and perturb samples
118     x_t = x * alphas_noise[..., None] + eps * sigmas_noise[..., None]
119
120     # w(t) coefficients
```

```python
            w = ((1 - alphas_noise) * sigmas_noise)[..., None]

            # compute score distillation update
            if method == 'l-vsd':
                xp = x.detach()
                for j in range(lora_steps):
                    r_optimizer.zero_grad()
                    q_muo = q_mu.detach()
                    q_sigmao = q_sigma.detach()
                    loss_r = F.mse_loss(q_muo, r_mu, reduction="sum") +
                        F.mse_loss(q_sigmao, r_sigma, reduction="sum")

                    loss_r.backward()
                    r_optimizer.step()

            with torch.no_grad():
                # \nabla \log p_t(x_t)
                score_p = calc_perturbed_gaussian_score(x_t, p_mu, p_sigma,
                    alphas_noise, sigmas_noise)

                if method == 'sds':
                    # -[\nabla \log p_t(x_t) - eps]
                    grad = -w * (score_p - eps)
                elif method == 'vsd':
                    # \nabla \log q_t(x_t | c) - centering trick
                    cond_mu = x.detach()
                    cond_sigma = torch.zeros_like(q_sigma)
                    score_q = calc_perturbed_gaussian_score(x_t, cond_mu,
                        cond_sigma, alphas_noise, sigmas_noise)

                    # -[\nabla \log p_t(x_t) - \nabla \log q_t(x_t | c)]
                    grad = -w * (score_p - score_q)
                elif method == 'real-vsd' or method == 'l-vsd':
                    cond_mu = r_mu.detach()
                    cond_sigma = r_sigma.detach()
                    score_q_appx = calc_perturbed_gaussian_score(x_t, cond_mu,
                        cond_sigma, alphas_noise, sigmas_noise)

                    grad = -w * (score_p - score_q_appx)

            # reparameterization trick for backpropagation
            # d(loss)/d(latents) = latents - target = latents - (latents - grad)
                = grad
            grad = torch.nan_to_num(grad)
            target = (x_t - grad).detach()
            loss = 0.5 * F.mse_loss(x_t, target, reduction="sum") / N_render

            loss.backward()
            optimizer.step()
            if scheduler is not None:
                scheduler.step()

            if method == 'real-vsd':
                r_mu_previous = r_mu.detach()
                r_sigma_previous = r_sigma.detach()
                xp = x.detach()
                for j in range(lora_steps):
                    r_optimizer.zero_grad()
                    q_muo = q_mu.detach()
                    q_sigmao = q_sigma.detach()
                    loss_r = F.mse_loss(q_muo, r_mu, reduction="sum") +
                        F.mse_loss(q_sigmao, r_sigma, reduction="sum")

                    loss_r.backward()
```

```
            r_optimizer.step()

    # logging
    if i % logging_steps == 0:
        state_dict.append({
            'step': i,
            'q_mu': q_mu.detach().cpu().numpy(),
            'q_sigma': q_sigma.detach().cpu().numpy(),
        })

        # save sample positions
        with torch.no_grad():
            p_samples = sample_gassian(p_mu, p_sigma, seed=vis_seed_true[0])
            p_samples = p_samples.detach().cpu().numpy()

            q_samples = sample_gassian(q_mu, q_sigma, seed=vis_seed[0])
            q_samples = q_samples.detach().cpu().numpy()

            if method == 'real-vsd':
                r_samples = sample_gassian(r_mu_previous, r_sigma_previous,
                    seed=vis_seed2[0])
                r_samples = r_samples.detach().cpu().numpy()
            else:
                r_samples = sample_gassian(r_mu, r_sigma, seed=vis_seed2[0])
                r_samples = r_samples.detach().cpu().numpy()

            vis_samples.append(np.stack([p_samples, q_samples, r_samples],
                0))
```

