# OpenReview forum: "Advancing Text-to-3D Generation with Linearized Lookahead Variational Score Distillation"
_ICLR.cc/2025/Conference — Submitted to ICLR 2025_

### Official Review · Reviewer_mA7z · 2024-10-30

**Soundness:** 2
**Presentation:** 2
**Contribution:** 2
**Rating:** 5
**Confidence:** 4

**Summary:**

The authors analyzed VSD by identifying critical gaps between its theory and implementation, and introduced L^2-VSD, an efficient and easily implemented variant to mitigate the mismatching issues. The author also showed that L^2-VSD integrates seamlessly with other VSD-based techniques, enhancing its practical utility.

**Strengths:**

This paper introduces a VSD variant addressing its practical training instability due to potential overfitting. The authors propose a lookahead scheme by updating LoRA before NeRF, contrary to VSD's approach. Additionally, they present a linearized strategy to mitigate the collapse caused by lookahead. The linearized strategy's formulation is clearly presented.

**Weaknesses:**

Figure placement is problematic. Figure 1, illustrating VSD's distribution alignment drawback, appears in Section 1 on page 2 but is explained in Section 3 on page 5. This forces readers to navigate back and forth to understand the figure's content.

The claim that few efforts focus on improving VSD is inaccurate. Besides ESD, which is examined in the paper, several other studies [1,2,3] have explored VSD variants, contradicting the authors' assertion.
The study is limited to NeRF+DmTet 3D representations. Score distillation is independent of 3D representation types. The authors should extend their experiments to include 3DGS-based methods, as demonstrated in works [4,5].

[1] Wei, Min, et al. "Adversarial Score Distillation: When score distillation meets GAN." Proceedings of the IEEE/CVF Conference on Computer Vision and Pattern Recognition. 2024.
[2] Lee, Kyungmin, Kihyuk Sohn, and Jinwoo Shin. "DreamFlow: High-quality text-to-3D generation by Approximating Probability Flow." The Twelfth International Conference on Learning Representations. 2024.
[3]  Ma, Zhiyuan, et al. "ScaleDreamer: Scalable Text-to-3D Synthesis with Asynchronous Score Distillation." European Conference on Computer Vision. 2024
[4] Tang, Jiaxiang, et al. "DreamGaussian: Generative Gaussian Splatting for Efficient 3D Content Creation." The Twelfth International Conference on Learning Representations.
[5] Liang, Yixun, et al. "Luciddreamer: Towards high-fidelity text-to-3d generation via interval score matching." Proceedings of the IEEE/CVF Conference on Computer Vision and Pattern Recognition. 2024.

**Questions:**

Why lookahead is important? In lines 227-229, the authors claimed that Lookahead-VSD optimizes NeRF faster than VSD for geometries and textures. However, Fig. 2b shows that Lookahead-VSD produces oversaturated textures compared to VSD (Fig. 2a), worsening as $\gamma$ increases from 1 to 5. This contradicts the authors' assertion that Lookahead is an effective strategy. The authors acknowledged this issue in lines 231-232 but do not explain the apparent contradiction.

The necessity of more LoRA steps. In Section 3, the authors extensively discussed the need for more LoRA steps (measured by $\gamma$) to better align with rendered image distributions. However, they neither incorporate $\gamma$ variation in the proposed methods nor include it in the ablation study, merely stating it is unnecessary. This raises questions: Is $\gamma$ not suitable for 3D generation? If so, why dedicate so much discussion to it?
Wrong CLIP similarity. The biggest puzzle for me is that lower CLIP similarity scores are considered better. Given that CLIP measures cosine similarity between text and image embeddings, higher scores should indicate better performance. While it is expected for VSD to have higher similarity than SDS, it is counterintuitive that L^2-VSD yields lower values.

---

> ### Author Response · Authors · 2024-11-22
>
> Thank you for your valuable and constructive review! We try to address your concern below.
>
> **W1. Problematic figure placement.**
>
> Thank you for your suggestion! But now it's hard for us to adjust this, and we would really like to reorganize the placement before the next version.
>
> **W2. Inaccuracy on related work. And should demonstrate on other representations like 3DGS.**
>
> Thank you for your reminder! We have included these papers and modified our description in the introduction. Also, we provide more results generated in the second and third stages in the appendix. The comparison of the geometry and texture refinement can prove the effectiveness of our method. Due to the time limit, we are very willing to demonstrate on 3DGS-based method like GaussianDreamer before the next version!
>
> **Q1: Lookahead will bring over-saturated then why lookahead is important.**
>
> Thanks for your question!
>
> Firstly, NeRF converges faster indeed in L-VSD than VSD. Secondly, we want to argue that in Figure 2b, it's obvious that the results have a much clearer shape with edges though still over-saturated. From the last figure of Figure 2b, we find that we can get better results by lowering the learning rate, then we come to the inference that "Lookahead" is the effective direction to investigate, while "lookahead" only is not effective enough. We also claim that only by lowering the learning rate is very unstable. We provide massive failed results in Figures 16 and 17 in the appendix. So lookahead is important and effective but not sufficient.
>
> **Q2: About the necessity of LoRA steps.**
>
> Thank you for pointing out that! Sorry for making Section 3.1 so confusing. We include Section 3.1 of investigating the impact of more LoRA steps in order to show a more complete map of possible factors. Also, as many statements in L-VSD are related to the convergence of LoRA, we believe including this part can help people have a better overview of the process. Now we have modified the section title to express more clearly.
>
>
> **Q3: Contradictive number with CLIP similarity.**
>
> Thank you for your careful review! We have a typo in the original paper. Actually, we calculate the arccos angle of the CLIP similarity. The smaller the angle is, the higher the original CLIP similarity is. We have modified this sentence about the evaluation metrics in the revised PDF.

---

> > ### Comment · Reviewer_mA7z · 2024-11-24
> >
> > Thanks for the response from the author. However, I still have concerns about the focus on repetitive LoRA finetuning. In your example in Figure 17, you demonstrate how this method, referred to as L-VSD, can lead to collapsed results. This raises doubts about the necessity of repetitive LoRA finetuning. Although you attempt to improve on the flawed L-VSD by progressing to L²-VSD, the results do not surpass VSD. This is evident in your additional examples. For example, Figure 19 indicates that the texture quality of L²-VSD is not superior to VSD, and in Figure 22, the results for "an exercise bike" produced by L²-VSD are even worse in terms of geometry compared to VSD. Therefore, I have decided to keep my original score.

---

> ### Author Response · Authors · 2024-11-24
>
> Thank you for your active response! But we still want to make some discussions here. We hope it can address your concern to some extent.
>
> 1. Actually the collasion of L-VSD doesn't lead to the conclusion of **repetitive LoRA finetuning**. L-VSD and our method takes the same LoRA finetuning times as VSD, once only. And we also design experiments in section3.3. to demonstrate that repetitive LoRA finetuning is unnecessary.
>
> 2. We don't focus on the repetitive LoRA finetuning, but we focus on the approximation distribution $q^{\mu}$ proposed by VSD, where the root of mismatching problem in VSD lies. Maybe you can refer to our answer to @LcCM W1.
>
> 3. We strongly refer you to additional results in Figure.8 with resolution of 256. In Figure.19,  VSD generates destroyed car with random red color, connecting destroyed car with a fire but our method generates more purely. And the texture of hand and the bowl in the bottom is also more realistic. In Figure.22, **for the limits of computation and high memory demand of high-order correction generation, we can only compare with high-order ablation with resolution of 64**. As we can observe, the results of our method generates more complete scene, and we strongly suspect that the geometry quality of this more complex scene is limited by the resolution.

---

### Official Review · Reviewer_w9Qs · 2024-10-30

**Soundness:** 2
**Presentation:** 2
**Contribution:** 2
**Rating:** 5
**Confidence:** 4

**Summary:**

This paper studies the issues of Variational Score Distillation (VSD) and proposes solutions to these problems. Specifically, the study identifies a gap between the theories and practical implementation of VSD. To bridge this gap, the first suggested approach is to repeat the LoRA training multiple times. The second approach is to run the LoRA training before the 3D model learning. However, the authors observe that simply combining these two methods does not lead to good performance. Furthermore, they find that the "training LoRA first" approach results in an unstable high-order term that negatively impacts generation quality. As a solution, they propose removing the high-order term from the update equation.

**Strengths:**

Strength:
1. The paper offers a solid analysis of VSD and proposes improvements based on this analysis.
2. The L2-VSD reduces computational overhead in the proposed methods.

**Weaknesses:**

Weaknesses:
1. The representation and organization of the paper in its current form are very confusing, especially in Section 3.1. This section discusses the importance of repeating the LoRA training multiple times, yet concludes that it is "somehow important but not sufficient." Additionally, this method is NOT included in the final equations (Eq 9 and Eq 11), raising questions about the section's relevance.

2. An important ablation study is missing: method with the high-order term vs. the proposed method vs. standard VSD. This makes it difficult to assess the importance of removing the high-order term for practical use cases.

3. The generated results using the proposed method still show over-saturated colors. This is visible in Fig. 7 (burger, fox, robot) and Fig. 8 (cake), indicating that the over-saturation issue has not been fully addressed.

4. VSD is known as one of the slowest SDS-like methods due to the optimization of LoRA. There are multiple methods that improve VSD's efficiency and achieve better results, such as CSD[1] and LODS[2], the proposed method is even slower than standard VSD, significantly limiting its practical utility.

5. Presentation inconsistency. In Fig. 7, the label unexpectedly changes to "L2-VSD".

[1] Text-to-3d with classifier score distillation, ICLR23
[2] Learn to optimize denoising scores for 3d generation, ECCV24

**Questions:**

Questions:

1. In addition to the weaknesses listed above, I feel that the proposed method still does not fully address the problems of VSD. Section 3.1 emphasizes the importance of "making the Lora model better converged", but I don't see how the lora model's convergence can be improved by the final Eq. 11.

2. Furthermore, if we consider an extreme case where the Lora model is perfectly trained, it would generate Gaussian noise identical to the noise added to the input. In this case, VSD would be reduced to standard SDS, which contradicts the intuition behind the proposed methods.

---

> ### Author Response · Authors · 2024-11-22
>
> Thank you for your valuable and constructive review! We try to address your concern below.
>
> **W1: Section 3.1 seems to be confusing, and irrelevant about the method.**
>
> Thanks for your valuable suggestion! Actually, we are trying to show the complete investigation maps. And in terms of discussing the impact of the convergence of "approximated distribution" in the original paper, we thought this section is necessary and helpful for a complete comprehensive study. We have modified the section title to express more clearly. Also, we changed the vibe of the conclusion of Section 3.1, to avoid confusing.
>
> **W2: Ablation experiments on the method with high-order terms are missing.**
>
> Thank you for your suggestion! We have done this experiment on another two samples and provided the results in Figure 22 in the appendix. We name the method HL-VSD (high-order lookahead VSD) because we use the high-order term instead of the linear term to correct the score. As shown in the figure, the results all collapse and become irrecognizable, which proves the effectiveness and necessity of linearized lookahead.
>
> **W3: Over-saturated problems with the generated results.**
>
> Yes, we don't claim that we have solved this problem completely, but we believe the results outperform VSD's results. Also, we provide more high-resolution results in the appendix like Figures 18, 19, and 22, and update the images in Figure 8. We believe these will strongly support our proposed method.
>
> **W4: The method consumes more time and other methods like CSD and LODS improves the efficiency.**
>
> Thank you for your reference!
>
> Firstly, CSD is not a VSD-based method. LODS changes the learning objective of VSD and deviates from the VSD-based method. But as the pipeline is similar, we believe our method can be incorporated into LODS too. We are willing to implement this before the next version.
>
> Secondly, we believe our method is orthogonal with other techniques that accelerate the whole process. As mentioned by YuaJ, we believe our method $L^2$-VSD, which is now implemented on VSD, can be transferred to these accelerated modern pipelines like GaussianDreamer as VSD and SDS, because SDS, VSD, and our $L^2$-VSD are all 3D distillation algorithms, orthogonal to the pipeline design.
>
> **W5: A typo in Fig.7**
>
> Thank you for your detailed review! We have corrected it.
>
> **Q1: Proposed method can't improve the LoRA convergence.**
>
> Firstly, through experiments in Section 3.1, we are trying to show that improving the convergence of LoRA is not the key factor. We include this part's discussion to provide a global view.
>
> Secondly, as written in Line 228, the loss of L-VSD with $\gamma=1$ floats in a similar level to VSD with $\gamma=10$, which suggests that lookahead in the LoRA model makes it faster to adapt to the current distribution, thus improving the convergence compared to VSD.
>
> **Q2: VSD reduced to standard SDS in an extreme case, which contradicts the intuition behind the proposed methods.**
>
> Thank you for your consideration of this subject!
>
> But firstly SDS is a special case of VSD when the particle number equals one. Even if we have a perfect LoRA model, as the timestep in LoRA training is different from the timestep in object training, it will only predict the mean of the added noise.
>
> Secondly, we don't emphasize that improving the LoRA convergence is the only key factor for the performance. Our experiments in Section 3.3 prove that overfitting in the LoRA model is harmful to the performance.

---

> > ### Comment · Reviewer_w9Qs · 2024-11-26
> >
> > Thanks for the response. My concerns are partly addressed in the rebuttal, but I am still concerned about the efficiency, and the visual results compared to other methods. Other reviewers also raised concerns about the visual results. I will keep my original rating.

---

> ### Author Response · Authors · 2024-11-26
>
> Thank you for the response!
>
> But we think our contribution is (1) finding out an important flaw with a representative method in score distillation 3D generation; (2) design extensive experiments to demonstrate the key factor; (3) propose the linearized-lookahead method to tackle this problem. We don't directly propose a whole new concurrent method compared with others.
>
> In terms of efficiency, we have responsed that by taking last-layer approximation, we can save more time. And our method is orthogonal to other acceleration methods. So we are very optimistic about the future of improment.
>
> More importantly, **about the visual results**. As we have provided more visual results in the revised PDF like figure.8,18,19 and 22, could you please tell us where or which samples do you have concern with? Also, could you please tell us which reviewer also raise concerns about the visual results in the revised PDF? Reviewer mA7z has concern with figure.22, which is mainly designed for comparison with high-order correction as you demand. We set the resolution as 64 because the huge memory it takes. And reviewer mA7z hasn't respond to us. Could you please share more opinions on it?
>
> As we have spared no effort trying to addressing your concern, we are eager to know what the problem still lies, rather than accepting "oh, someone else also says that", which is not supported with a conclusion of a discussion. Thank you again.

---

### Official Review · Reviewer_MLn6 · 2024-11-03

**Soundness:** 2
**Presentation:** 2
**Contribution:** 2
**Rating:** 3
**Confidence:** 4

**Summary:**

The paper introduces Linearized Lookahead Variational Score Distillation (L2-VSD) to improve the convergence of VSD for improved text-to-3D generation with text-to-image diffusion models. The paper aims to improve the NeRF optimization using VSD, by first analyzing the convergence behavior of LoRA layer, and suggest a Lookahead score distillation which can reduce the gradient norm. The author states that this method can better adjust the gradient information based on current 3D model and can improve the stability of 3D optimization as well as the quality of 3D generation. The provided experiments compare with VSD with its variant, and show improvements.

**Strengths:**

1. L2-VSD provides more stable convergence by utilizing a linearized lookahead correction.
2. L2-VSD can be incorporated into other VSD-based frameworks, such as HiFA.

**Weaknesses:**

The visual results presented for L2-VSD do not clearly demonstrate an improvement over those generated by VSD. Specifically, Figure 7 does not convincingly address known issues with VSD, such as saturated colors and visual artifacts in 3D assets. Additionally, the quantitative results in Table 1 are somewhat misleading. In most contexts, a higher CLIP similarity score indicates a better match with the prompt, yet L2-VSD, which has a lower CLIP similarity than other methods, is highlighted in bold. The evaluation is also based on 20 prompts with 120 views, which could lead to a saturated distribution, raising concerns about the reliability of FID as a measure of the proposed method’s quality.

Regarding Figure 3, the observed reduction in loss is likely expected, given that the random timestep selection during optimization naturally reduces variance. The authors attempt to correlate this reduction in loss with improved 3D generation quality by showing a single example in Figure 2. However, given the inherent randomness in generative models, drawing meaningful conclusions from a single example is not advisable and may not support a causal relationship.

The paper’s clarity could also benefit from further refinement, as the core argument is challenging to follow. The main points seem to be: 1) fitting the LoRA model in VSD with additional optimization steps can improve 3D generation quality; 2) however, a naive application of this approach did not yield satisfactory results; and 3) therefore, the authors propose a linearized VSD with a second-order approximation to improve convergence. This reasoning is presented in a somewhat disjointed manner, making it difficult to follow the logic and understand the proposed method’s effectiveness over existing approaches. Additionally, the results presented do not appear to fully support the claims made.

**Questions:**

1. Could the author justify the quantitative results in Table 1?
2. Does the proposed method can be applied to mesh fine-tuning (e.g., stage 2 of Prolific dreamer)?

---

> ### Author Response · Authors · 2024-11-22
>
> Thank you for your valuable and careful review! We try to address your concern below.
>
> **W1. The visual results presented for L2-VSD do not clearly demonstrate an improvement over those generated by VSD. And the CLIP similarity number is misleading.**
>
> Firstly, we have provided more high-resolution results in Figures 8, 18, 19, and 22 in the appendix, especially by Figure.8 which is generated with high-resolution 256. The blurry degration of performance in Figure.22 is caused due to low-resolution 64. Figure.22 is provided mainly for comparison with high-order generation, and this process demands high memory. We believe these results can help us demonstrate the effectiveness of our method.
>
> Secondly, sorry for the incomplete metrics description! Actually, we calculate the arccos angle of the usual defined CLIP similarity, so the lower the angle is, the higher the CLIP similarity. We have corrected our description of evaluation metrics in the revised PDF. Besides, we calculate the FID averaged across different prompts while the FID of each prompt is measured across 120 views spanning 360°.
>
> **W2. Drawing meaningful conclusions from loss reduction in Fig. 3 and improved quality in Fig. 2a is not advisable.**
>
> Thank you for your question! But we never correlate the reduction in loss with the quality improvement. In the original paper, we want to ensure the convergence of LoRA by the loss curve of Figure 3. Besides, as written in Section 3.1 in the original paper, the overall quality of Figure 2a "does not" witness a continual improvement. So we come to the conclusion that better convergence is not sufficient and not the key factor.  And that's also why we claim that investigating in lookahead is more important.
>
> **W3. About the clearness of clarity.**
>
> Thank you for your summarization! Our main points are 1) analyzing two possible factors that cause the gaps of VSD and justify them separately, 2) ensuring the convergence of LoRA is not that helpful and increases the computation budget hugely, 3) correcting the optimization order of VSD has the potential to bring better results, but naively using it doesn't help, and 4) combining the stable but less satisfying VSD and unstable but possibly better L-VSD together and propose the $L^2$-VSD. We recognize the convergence part of Section 3.1 is kind of misleading and have modified the section title! We will try our best to make a better statement to catch the logic.
>
> **Q1: Justify the quantitative results in Table.1**
>
> The answer is included in the answer to W1.
>
> **Q2: Can the method be applied to mesh fine-tuning?**
>
> Yes! Our method can generalize to other representations. We conduct experiments in stages 2 and 3 in VSD. And we have provided the results in Figures 18 and 19 in the appendix. This comparison reflects the generalization of our method on other representations. We are willing to demonstrate on 3DGS before the next version.

---

### Official Review · Reviewer_LcCM · 2024-11-03

**Soundness:** 2
**Presentation:** 2
**Contribution:** 2
**Rating:** 3
**Confidence:** 4

**Summary:**

This work empirically analyzes the two theory-implementation gaps of Variational Score Distillation (VSD): (1) Convergence of the LoRA model; (2) Misalignment of LoRA model with current 3D model. Based on the analysis, this work further introduces L-VSD and L$^2$-VSD, both of which are lookahead variants of VSD and enable the LoRA model to be aligned with the current 3D model to match theoretical derivations. In particular, L$^2$-VSD is an interpolation of VSD and L-VSD, obtained by discarding the high-order Taylor expansion terms of L-VSD. For evaluation, this work provides qualitative and quantitative comparisons of L-VSD and previous methods: SDS, VSD, ESD, and HiFA.

**Strengths:**

1. Variational Score Distillation (VSD) is a representative score distillation method for diffusion-guided 3D generation. The defect analysis and improvement of VSD may provide inspiration for subsequent research. To the best of my knowledge, the analysis of LoRA training for VSD is original and somewhat interesting.

2. The paper is well-written and clearly structured.

**Weaknesses:**

1. This work only explores the potential issues of LoRA training for VSD, which limits the scope of this work. In fact, VSD is just one of the score distillation methods, and some newer SDS techniques such as ISM do not require a LoRA model. Even for VSD, the introduction of LoRA is already a compromise in implementation, and further analysis of its theory-implementation gaps seems trivial. I'm not opposed to this kind of exploration, but I expect it to bring more significant results than the marginal improvements presented in the paper (e.g., Figure 7). Moreover, one of the shortcomings of VSD is the time-consuming training, and L$^2$-VSD requires 43% additional training time compared to VSD (as reported in Table 2).

2. I notice that most of the analysis (e.g., Figure 2, Figure 3, Figure 4) on VSD are based on a single example "a delicious hamburger", which weakens the generalizability of the results. For quantitative analysis like Figure 3 and Figure 4, experiments should be performed on multiple samples to ensure generalizability and reduce noise.

3. In Section 3.1, the authors point out that LoRA's finite step optimization is under-convergence and deviates from the theoretical derivation of VSD. However, in Section 3.3 and Section 4, the authors claim that the lookahead mechanism leads to the risk of over-convergence and a linear lookahead mechanism is needed to avoid overfitting. There seems to be a lack of motivation to explain why LoRA overfitting is harmful, because it is obviously consistent with the theory of VSD.

4. It is difficult to understand "the necessity for the LoRA model to look ahead" claimed in Line 230. From the results in Figure 2, L-VSD does not seem to have any advantage over VSD when $\gamma$ = 1, 2, and 5.

5. Figure 1 is unclear and confusing. All I can understand from Figure 1 is that the L-VSD fits better than the VSD, but it is not clear what the role of the LoRA distribution r(x) is in this figure.

**Questions:**

1. For L-VSD, in a certain iteration step $i$, are the input $x_{t'}$ used for LoRA training (Eq.(6)) and the input $x_t$ used for NeRF training (Eq.(5)) sampled from the same camera view? Do they use different timesteps?

2. In Figure 2(b), it seems that lowering the learning rate of LoRA is quite effective for L-VSD. So why do we need to use the computationally expensive L$^2$-VSD?

---

> ### Author Response · Authors · 2024-11-22
>
> Thank you for the valuable and constructive review! We try to address your concern and argue for the meaning of our work below. Hope to discuss more if you have contrasting opinions!
>
> **W1: This work seems to only explore the LoRA training of VSD, which limits the scope of this work. The introduction of LoRA in VSD is already a compromise in implementation, and it doesn't make sense to further analyze its theory-implementation gaps. Also, the improvements seem to be marginal with respect to the additional time.**
>
> Thank you for your consideration on this meaningful subject! But we might have different thoughts on it.
>
> 1. Firstly, as Reviewer LcCM has mentioned in "Strengths", VSD is a representative score distillation method for 3D generation and has shown a powerful potential to generate more vivid assets. We can basically divide modern 3D score distillation methods into VSD-based and SDS-based ones, and SDS is actually a subclass of VSD with a single particle. So we believe efforts in analyzing these basic frameworks are meaningful for proposing a new framework.
>
> 2. Secondly, our motivation is to investigate the gaps in VSD though it already establishes a mature and complete theoretical framework. One of the biggest differences between VSD and SDS is **introducing the approximated score of 3D views**. So we don't mean to **"only explore" the LoRA training** for VSD, but **have to explore this "approximated score"**, which is expressed within LoRA.
>
> 3. Lastly, **we want to correct seriously that "the implementation of VSD is never a compromise but a default recommendation"**. We strongly refer you to Section 3.2 of the original paper of VSD. Using LoRA rather than a small U-Net can greatly enhance performance. Also, the introduction of LoRA is not the root cause of the gaps we analyze.
>
> **W2: Figures 2,3, and 4 use the same prompt case, which weakens the generalizability of the results. For qualitative analysis, experiments should be performed on multiple samples.**
>
> Thank you for pointing out that! Actually, we intended to offer a more comprehensive comparison from the global view, so we use the hamburger case as the global illustrative example. To address your concern and provide more convincing signals, we included the average loss curves with multiple samples in Figure 20b of the revised PDF. The conclusion holds the same as in Section 3.1. Also, we provide one sample "crown" other than "hamburger" to augment the proof.
>
> **W3: Why LoRA overfitting is harmful, which is obviously consistent with the theory of VSD.**
>
> Thank you for your interest. As written in Line 268 in Section 3.3, from the theory of VSD, LoRA is used to estimate the current distribution of 3D views. But current training of LoRA is to overfit on a specific view of a specific possible object. So the distribution is biased. Moreover, overfitting the LoRA model will destroy the prior information with respect to text from Stable Diffusion, which is harmful to the performance evidenced by the original paper of VSD.
>
> **W4: Why is it necessary for the LoRA model to look ahead from the results in figure2, that L-VSD does not seem to have any advantage over VSD?**
>
> From our observation, we find the geometry edges of these results in Figure 2b are more clear and have a more complete shape. Also as mentioned in Line 228 in Section 3.2, other than visual judgment, we find that the loss level of L-VSD with $\gamma=1$ floats in a similar level to VSD with $\gamma=10$, which suggests that L-VSD is better for fit on the current distribution.
>
> **W5: What is the meaning of $r(x)$ in Figure 1?**
>
> We are sorry for the confusion. But actually, we want to show the corrective guidance from $r(x)$. It's kind of hard to draw an arrow between distributions of different steps.
>
> **Q1: For L-VSD, in a certain iteration step $i$, are the input $x_{t'}$ used for LoRA training (Eq.(6)) and the input $x_t$ used for NeRF training (Eq.(5)) sampled from the same camera view? Do they use different timesteps?**
>
> Yes! They are sampled from the same camera view, which is held the same as in VSD. But they use different timesteps, which are also held as the same as in VSD.
>
> **Q2: Lowering the learning rate of LoRA seems to be quite effective, so why do we need $L^2$-VSD?**
>
> Thank you for your question! As shown in Figure 16 and Figure 17, the strategy of lowering the learning rate is very unstable and uncontrollable. We provide many failure cases of L-VSD in the appendix. Also, we are motivated by this positive sample, believing that a lookahead for the LoRA model is necessary. And then investigate more on the difference between VSD and L-VSD, thus having more stable and controllable $L^2$-VSD.

---

### Official Review · Reviewer_YuaJ · 2024-11-04

**Soundness:** 3
**Presentation:** 3
**Contribution:** 3
**Rating:** 6
**Confidence:** 4

**Summary:**

This paper addresses the text-to-3D generation problem using the variational score distillation (VSD) method. The authors identify issues in the practical implementation of VSD, specifically a mismatching problem between the LoRA and 3D distributions. They propose the Linearized Lookahead Variational Score Distillation ($L^2$-VSD) method to improve the generation quality. The paper conducts in-depth analyses and experiments, comparing $L^2$-VSD with other baseline methods and demonstrating its superiority in terms of generation quality and compatibility with other techniques.

**Strengths:**

+ The paper is well-structured. The writing is clear. The methodology section presents a detailed comparison between VSD and L-VSD, followed by a clear derivation of $L^2$-VSD.
+ The proposed $L^2$-VSD method addresses the mismatching problem in VSD by adjusting the optimization order and using a linearized variant for score distillation.

**Weaknesses:**

- The proposed $L^2$-VSD method seems to be highly dependent on the specific settings and assumptions of the VSD framework. It is not clear how well it would generalize to other text-to-3D generation methods like Gaussian Dreamer or LucidDreamer, which are recent SOTA, or different types of 3D representations such as Gaussian Splatting. As the original VSD takes 8 hours to generate one 3D model, while Gaussian Dreamer takes only 15 mins. The reviewer is afraid the proposed method does not give significant contribution to the current literature.
- The definition of the mismatching problem between LoRA and 3D distributions could be more precise. Besides, is there any metric to estimate the degree of mismatching?
- Figure 2a: In the original VSD results, as γ increases, the "hamburger" depicted in the figure appears smaller. However, the manuscript states that "the shape of the hamburger becomes more reasonable and clearer as γ rises," which is confusing and requires clarification.
- Figure 2b: Although γ is increased, the learning rate is decreased. These two adjustments seem to counteract each other. Initially, the LoRA learning rate is relatively high, allowing more to be learned in a single update step. With the reduced learning rate, the amount updated in each step decreases, but the number of update steps (γ) increases. This appears to result in a balancing effect, and the manuscript should address this interaction.
- Figure 3: I am particularly interested in the loss variations during the initial few dozen steps for the four curves presented. Could the authors provide the relevant data to illustrate the loss behavior in the early training stages?
- Figure 4: Would it be possible to include the curve for the VSD method in this figure for comparative purposes?
- Last-Layer Approximation: The manuscript mentions that using the last-layer approximation can eliminate floaters but leads to a reduction in quality. Could the authors elaborate on why this trade-off occurs?
- Final Methodology: In the proposed final approach, is it still necessary to use γ to increase the number of LoRA training iterations? Clarification on the necessity and impact of γ in the final method would be beneficial.
- Effectiveness of Quality Improvement: Based on the current set of result images, it is challenging to ascertain whether the quality has been effectively enhanced. Could the authors provide additional result images to better demonstrate the improvements?
- Scope of VSD Improvements: The paper appears to implement improvements only in the first phase of VSD. Is this the final outcome, or are there plans to apply similar geometry and texture improvements in the second and third phases of VSD? Clarification on whether subsequent phases will also be addressed is needed.

**Questions:**

Please see the weakness.

---

> ### Author Response · Authors · 2024-11-22
>
> Thank you for the valuable and careful review! We do appreciate your careful reading. Below, we try to address your concern point by point.
>
> **Q1&Q10: How well can the method generalize to other methods like GaussianDreamer, different 3D representation, or subsequent training phase? How to mitigate the long training time problem?**
>
> 1. In terms of 3D representations and method, from the aspect of algorithm, our method $L^2$-VSD can generalize as SDS and VSD as long as the rendering process of the chosen representation is differentiable. But we do acknowledge that there might exist biases between different representations. So we provide more results in meshes from the second and the third stages in Fig.18 and Fig.19. The results indicate that our method can still generate more realistic results in mesh. Due to time limitations, we promise to experiment on 3DGS before the next version.
>
> 2. We agree with Reviewer YuaJ that the computation time of VSD is huge, and now many concurrent methods like GaussianDreamer exist to accelerate the whole process. We believe our method $L^2$-VSD, which is now implemented on VSD, can be transferred to these accelerated modern pipelines as VSD and SDS, because SDS, VSD, and our $L^2$-VSD are all 3D distillation algorithms, orthogonal to the pipeline design.
>
> **Q2: Could the mismatching problem be more precise? Are there any metrics?**
>
> Thank you for the valuable question. Mathematically, according to the equation.7 in the original paper of ProlificDreamer, the update rule of VSD follows the following ODE:
> $$
> \frac{d\theta_{\tau}}{d\tau} = -E_{t,\epsilon,c}[w(t)\left(
> -\sigma_t\nabla_{x_t}\log p_t(x_t|y^c) - (-\sigma_t\nabla_{x_t}\log q_t^{\mu_\tau}(x_t|y,c))
> \right)\frac{\partial g(\theta_\tau,c)}{\partial\theta_\tau}]
> $$
>
> where $q_t^{\mu_\tau}$ is the corresponding noisy object renderings distribution at diffusion time t w.r.t $\mu_\tau$ at ODE time $\tau$.
>
> In this sense, the real update rule of VSD actually uses $q_t^{\mu^{\tau-1}}$, which causes the mismatching problem. So maybe $D_{KL}(q_t^{\mu_\tau}(x_t|y,c)|q_t^{\mu_{\tau-1}}(x_t|y,c))$ can be used to precisely describe this problem. What is interesting is that, from this perspective, one "possible" solution is to penalize the change of $\mu_\tau$, but definitely resulting in much longer time.
>
> If you think this is helpful, we would be very glad to add this analysis into the final version.

---

> > ### Author Response · Authors · 2024-11-22
> >
> > **Q3: Description in Figure 2a requires clarification.**
> >
> > Thank you for pointing out that. From our observation, we think the overall geometry shape is getting more complete, symmetric, and reasonable. But we do agree with you on that it's kind of subjective to make such a conclusion. We have modified this statement in the revised PDF. And we believe this modification of the statement doesn't influence the logic, still supporting our claim that "improving the convergence of LoRA is not sufficient". Also, we provide another illustrative case in Figure 21.
> >
> > **Q4: The setting in Figure 2b seems to include factor interaction?**
> >
> > Actually, we don't decrease the learning in the first three sub-images, and only decrease the last image. So there is no interaction of these two variables.
> >
> > **Q5: Could the author provide loss data in the early training stages?**
> >
> > Thanks for your interest! We have provided the loss data in the appendix, Figure 20a. As shown by the curve, the loss is at a similar level at the start of distillation, which is probably because the objects haven't formed into a clear shape yet. So the predicted noises are all likely to be Gaussian.
> >
> > **Q6: Would it be possible to include the curve for the VSD method in Figure 4?**
> >
> > Sure! We have included the VSD loss curve in Fig.4. As we can observe, as long as the shape is within the normal range, the LoRA loss of L-VSD is comparable with VSD.
> >
> > **Q7: Could the author elaborate on why using last-layer approximation brings trade-off?**
> >
> > As the correction term is calculated with the Jacobian matrices of the LoRA model, if only use the last-layer gradients to approximate the correction, we actually ignore the information within other layers, thus losing accuracy.
> >
> > **Q8: Is it still necessary to use $\gamma$ to increase the number of LoRA training iterations?**
> >
> > Thank you for your suggestion! Through investigation of section 3.1, we believe it's not necessary to increase the LoRA training times as it doesn't help in VSD and we haven't seen improved quality. We also modified the section title to express more clearly.
> >
> > **Q9: Could the authors provide additional result images?**
> >
> > Yes. We have included more results shown in Figures 8, 15, 18, 19, and 22.

---

### Author Response · Authors · 2024-11-22
**Global Response**

We thank the reviewers for their insightful comments and valuable feedback. We are pleased that the reviewers found our paper to be **well-written, clearly structured, with a clear derivation of $L^2$-VSD** (@YuaJ, @LcCM), our analysis of VSD is **solid, original and somewhat interesting** (@w9Qs), and our perspective **may provide inspiration for subsequent research** (@LcCM)
***
We first summarize updates of the revised manuscript, followed by responses to individual comments. All revisions are highlighted in **red** in the updated version:

- Related Work (@mA7z; Line 053): Added references for VSD improvement and modified the sentence.
- Misleading Paragraph (@w9Qs, @mA7z; Line 192, 216, 248): Modified the section title to express more clearly.
- Description of Figure 2a (@YuaJ; Line 213): Modified into more objective description.
- Details of CLIP similarity (@MLn6, @mA7z; Section 5.3): Corrected the exact description of evaluation metrics.

**Also we add numerous experiments**
- Generalization of Figure 2 and Figure 3 (@LcCM; Appendix C.3): Provided averaged loss curve on multiple samples and another illustrative case.
- Figure 4 (@YuaJ; Appendix C.3): Added norm of VSD LoRA loss in Figure 4 for comparative purposes.
- More qualitative results (Figure 8, Figure 18, Figure 19, Figure 22): Included more results for comparison, especially Figure.8.
- Generalization on other representations (@YuaJ, @MLn6; Figure 18, Figure 19): Provided results on the second and third stages results.
- High-order term comparison (@w9Qs; Figure 22): Provided results with resolution 64 by using the high-order term only to correct.

---

### Meta-Review · Area_Chair_fiFy · 2024-12-22

**Metareview:**

Although one of the reviewers suggested acceptance, others raised concerns about the limited novelty and insufficient analysis. Specifically, focusing solely on the potential issues of LoRA training for VSD narrows the scope of this work. Additionally, the performance improvement over the baseline was marginal. The AC has reviewed the paper, the reviews, and the rebuttal and agrees that the current version is not ready for acceptance. The AC strongly encourages the authors to address all the reviewers' concerns and resubmit to a future venue.

**Additional Comments On Reviewer Discussion:**

There were no reviewer discussion.

---

### Decision · Program_Chairs · 2025-01-22

Reject